# Low pressure reversibly driving colossal barocaloric effect in two-dimensional vdW alkylammonium halides

Yi-Hong Gao[1,2,8], Dong-Hui Wang[3,8], Feng-Xia Hu ●[1,2,4] ✉, Qing-Zhen Huang[5,6], You-Ting Song[1,2], Shuai-Kang Yuan[1,2], Zheng-Ying Tian[1,2], Bing-Jie Wang[1,2], Zi-Bing Yu[1,2], Hou-Bo Zhou[1,2], Yue Kan[1,2], Yuan Lin[1,2], Jing Wang ●[1,2] ✉, Yun-liang Li ●[1,2,4] ✉, Ying Liu[3], Yun-Zhong Chen ●[1,2], Ji-Rong Sun ●[1,2,4], Tong-Yun Zhao[1,7] & Bao-Gen Shen[1,2,5] ✉

Plastic crystals as barocaloric materials exhibit the large entropy change rivalling freon, however, the limited pressure-sensitivity and large hysteresis of phase transition hinder the colossal barocaloric effect accomplished reversibly at low pressure. Here we report reversible colossal barocaloric effect at low pressure in two-dimensional van-der-Waals alkylammonium halides. Via introducing long carbon chains in ammonium halide plastic crystals, two-dimensional structure forms in $(CH_3-(CH_2)_{n-1})_2NH_2X$ (X: halogen element) with weak interlayer van-der-Waals force, which dictates interlayer expansion as large as 13% and consequently volume change as much as 12% during phase transition. Such anisotropic expansion provides sufficient space for carbon chains to undergo dramatic conformation disordering, which induces colossal entropy change with large pressure-sensitivity and small hysteresis. The record reversible colossal barocaloric effect with entropy change $\Delta S_r \sim 400\,J\,kg^{-1}\,K^{-1}$ at 0.08 GPa and adiabatic temperature change $\Delta T_r \sim 11\,K$ at 0.1 GPa highlights the design of novel barocaloric materials by engineering the dimensionality of plastic crystals.

On the continuous booming of resource-lack and environmental decay of global warming, the present cooling technique, mainly based on freon vapor compression system, needs to be substituted with efficient and zero-carbon emission cooling technique. Now leveraging the solid-state caloric effects for refrigeration has been intriguing alternative to conventional cooling technique, which takes advantages of zero direct carbon emission, high energy efficiency and susceptibility to integration[1]. Caloric effects are exhibited in materials where phase transition with thermal change can be induced by external fields, such

as magnetic, electric and mechanical field. Specifically, the magnetic and electric fields can drive magnetocaloric and electrocaloric effect, respectively[2–7]. And the mechanical fields involving uniaxial stress and hydrostatic pressure drive elastocaloric[8,9] and barocaloric effect respectively. The barocaloric cooling system superiorly features the accessible mechanical field, instead of costly magnetic field, and general barocaloric materials not requesting rigorous mechanical properties, which however is necessary for electrocaloric, magnetocaloric and elastocaloric materials[10,11]. Escaping from the restriction of

[1]Beijing National Laboratory for Condensed Matter Physics, Institute of Physics, Chinese Academy of Sciences, Beijing 100190, PR China. [2]School of Physical Sciences, University of Chinese Academy of Sciences, Beijing 101408, PR China. [3]College of Chemistry, Beijing Normal University, 100875 Beijing, PR China. [4]Songshan Lake Materials Laboratory, Dongguan, Guangdong 523808, PR China. [5]Ningbo Institute of Materials Technology & Engineering, Chinese Academy of Sciences, Ningbo, Zhejiang 315201, PR China. [6]Spallation Neutron Source Science Center, Dongguan 523803, PR China. [7]Ganjiang Innovation Academy, Chinese Academy of Sciences, Ganzhou, Jiangxi 341000, PR China. [8]These authors contributed equally: Yi-Hong Gao, Dong-Hui Wang. ✉e-mail: fxhu@iphy.ac.cn; wangjing@iphy.ac.cn; yunliangli@iphy.ac.cn; shenbg@iphy.ac.cn

mechanical property requirement and specific ferroicity such as ferromagnetism, ferroelectricity or ferroelasticity, barocaloric materials include a mass of structural phase change materials; developing from the incipient metallic compounds[12–19], some emergent barocaloric materials can exhibit colossal caloric effect[20–29] with entropy change larger than 100 J kg$^{-1}$ K$^{-1}$, the phase transitions of which relate to the coupling between the remarkable structural change and large order-disorder transition of order parameters such as molecular orientation order[20,21,23,25], conformational order of organic chain[27–29] and spin crossover[26].

Specifically, as a milestone in barocaloric cooling field, colossal caloric effect was achieved in polyalcohol plastic crystals[20], the entropy change during phase transition attaining from 384 J kg$^{-1}$ K$^{-1}$ in NPG ((CH$_3$)$_2$C(CH$_2$OH)$_2$) to 682 J kg$^{-1}$ K$^{-1}$ in TRIS ((NH$_2$)C(CH$_2$OH)$_3$), which approaches that of present freon refrigerant. However, the fatal low pressure-sensitivity and large hysteresis of phase transition (Table 1) bring about the irreversibility of caloric effect at low applying pressure, i.e., 0.1 GPa, in such three-dimension (3D) cubic plastic crystals, where the representative NPG exhibits none of reversible entropy change at pressure of 0.1 GPa. Underlyingly, the cubic plastic crystals maintain tight 3D structure via hydrogen bonds at ordered state[30], which limits the volume change also consequently pressure-sensitivity of phase transition. Meanwhile, large geometry incompatibility of two phases during phase transition related to the drastic symmetry breaking/restoration of lattice tends to cause large hysteresis[31–33]. Therefore, it is crucial to regulate comprehensively the entropy change, pressure sensitivity and hysteresis of phase transition, for achieving the reversible colossal barocaloric effect (BCE) driven by low pressure. It remains a challenge for BCE to enhance the reversibility while maintaining the magnitude as large as polyalcohol plastic crystals.

Here, we demonstrate that by introducing the alkyl chains in ammonium halide plastic crystal and constructing the two-dimensional (2D) van-der-Waals (vdW) alkylammonium halides, the colossal BCE with entropy change ~400 J kg$^{-1}$ K$^{-1}$ can be reversibly driven by pressure lower than 0.1 GPa. Single crystal x-ray diffraction (SC-XRD) indicated that long organic chains are assigned parallelly and anchored at the both ends of N−Cl plane to form the 2D structure in didecyl ammonium chloride (CH$_3$−(CH$_2$)$_9$)$_2$NH$_2$Cl (Fig. 1a). Within the 2D layered system maintained by intralayer N−H...Cl hydrogen bond and interlayer vdW interactions, the structural change across phase transition and relevant dynamics at molecular level were revealed by powder x-ray diffraction (PXRD), molecular dynamics (MD) simulation and temperature-variable infrared (IR) spectroscopy.

Our results demonstrated that the 2D (CH$_3$−(CH$_2$)$_9$)$_2$NH$_2$Cl undergoes interlayer expansion as large as 13% along the length direction of carbon chains (c axis) while ab plane remains nearly unchanged, consequently volume expansion as much as 12% occurs during phase transition. Such anisotropic expansion of lattice provides sufficient space for carbon chains to undergo dramatic conformation disordering, which induces colossal entropy change with large pressure-sensitivity and small hysteresis. MD simulation elucidated the specific disordering process from perspective of radial distribution function g(r) and dihedral angle of carbon chains, and the measured IR spectra gave fingerprint information across phase transition for each component, i.e., the NH$_2$$^+$ and N−H...Cl hydrogen bonds, the CH$_2$ and C−C chains, and the CH$_3$ tails, in the entire ((CH$_3$−(CH$_2$)$_9$)$_2$NH$_2$)$^+$ chains. The peculiar dynamics enables realization of colossal entropy change from disordering of organic chains, high sensitivity to pressure from large volume expansion, and relatively low hysteresis stemming from the low-dimension structure related low energy barrier of phase transition, thus accomplishing the record reversible colossal barocaloric entropy change $\Delta S_r$ ~ 400 J kg$^{-1}$ K$^{-1}$ at 0.08 GPa in (CH$_3$−(CH$_2$)$_9$)$_2$NH$_2$Cl affirmed by pressure differential scanning calorimetry. In addition, adiabatic temperature change $\Delta T_{ad}$ as large as 11 K at applying pressure of 0.1 GPa was demonstrated by direct barocaloric measurement. These performances exceed those of all other barocaloric materials reported to date (Table 1).

## Results
### 2D vdW crystalline structure and thermal properties of phase transition

The dialkyl ammonium halides (CH$_3$−(CH$_2$)$_{n-1}$)$_2$NH$_2$X (X: halogen element) evolve from the substitution of two alkyl chains (CH$_3$−(CH$_2$)$_{n-1}$) for two hydrogen atoms in ammonium halide molecule, in such process,

**Table 1 | Comparisons of colossal barocaloric materials with solid-solid phase transition exhibiting entropy change larger than 100 J kg$^{-1}$ K$^{-1}$**

| Compounds | $T_s$ (K) | $|\Delta S|$ (J kg$^{-1}$K$^{-1}$) | Hys. (K) T rate (K min$^{-1}$) | $dT_s/dP$ (K GPa$^{-1}$) | $\Delta V$ (E-5 m$^3$ kg$^{-1}$) | $|\Delta S_p|$ −0.1 GPa (J kg$^{-1}$K$^{-1}$) | $|\Delta S_r|$ −0.1 GPa (J kg$^{-1}$K$^{-1}$) | Refs. |
|---|---|---|---|---|---|---|---|---|
| dC$_{10}$Cl | 325 | 400 | 8 (1) | 190 | 11.9 | 400 | 400 | This work |
| NPG (CH$_3$)$_2$C(CH$_2$OH)$_2$ | 313 | 384 | 14 (0.1) | 133 | 4.6 | 0 | 0 | 20,21 |
| PG (CH$_3$)C(CH$_2$OH)$_3$ | 354 | 485 | 4 (2–4) | 79 | 5.1 | 445 | 155 | 23,55 |
| TRIS (NH$_2$)C(CH$_2$OH)$_3$ | 407 | 682 | 75 (2–4) | 37 | 3.7 | 0 | 0 | 23,56 |
| AMP (NH$_2$)(CH$_3$)C (CH$_2$OH)$_2$ | 353 | 632 | - | 64 | 4.6 | 0 | 0 | 23,55 |
| NPA (CH$_3$)$_3$C(CH$_2$OH) | 232 | 204 | 20 (2–4) | 220 | - | 0 | 0 | 23 |
| 1-Cl-ada | 254 | 132 | 9 (2–4) | 270 | 4.7 | 170 | 160 | 25 |
| 1-Br-ada | 308 | 102 | 9 (2.5–3) | 333 | 4 | 150 | 135 | 25 |
| 1-adamantanol | 361 | 210 | 15 | 179 | 4.5 | ~250 | 0 | 57 |
| 2-methyl-2-adamantanol | 375 | 371 | 11 | 241 | 8.8 | ~380 | ~350 | 57 |
| CaF$_2$ | 1400 | 226 | - | - | - | | | 22 |
| (CH$_3$-(CH$_2$)$_8$-NH$_3$)$_2$MnCl$_4$ | 294 | 212 | 5.2 (1) | 172 | - | 212 | 212 | 28 |
| (CH$_3$-(CH$_2$)$_9$-NH$_3$)$_2$MnCl$_4$ | 312 | 230 | 4.0 (1) | 150 | - | 230 | 230 | 28 |
| PEG10000/PET15000 | 334 | 426 | 21.8 (1) | 97 | - | 416 | 35.6 | 58 |
| NH$_4$SCN | 364 | 129 | ~25 | 300 | −3.8 | 129 | ~100 | 59 |

Parameters are listed containing the phase transition temperature during heating ($T_s$), entropy change during phase transition at atmosphere pressure ($|\Delta S|$), hysteresis (Hys.), the dependence of $T_s$ on pressure ($dT_s/dP$), volume change of phase transition from experiment ($\Delta V$), barocaloric entropy change induced by 0.1 GPa ($|\Delta S_p|$ −0.1 GPa) and reversible barocaloric entropy change induced by 0.1 GPa ($|\Delta S_r|$ −0.1 GPa). The ones from NPG to 2-methyl-2-adamantanol on the upper part denote 3D plastic crystals.

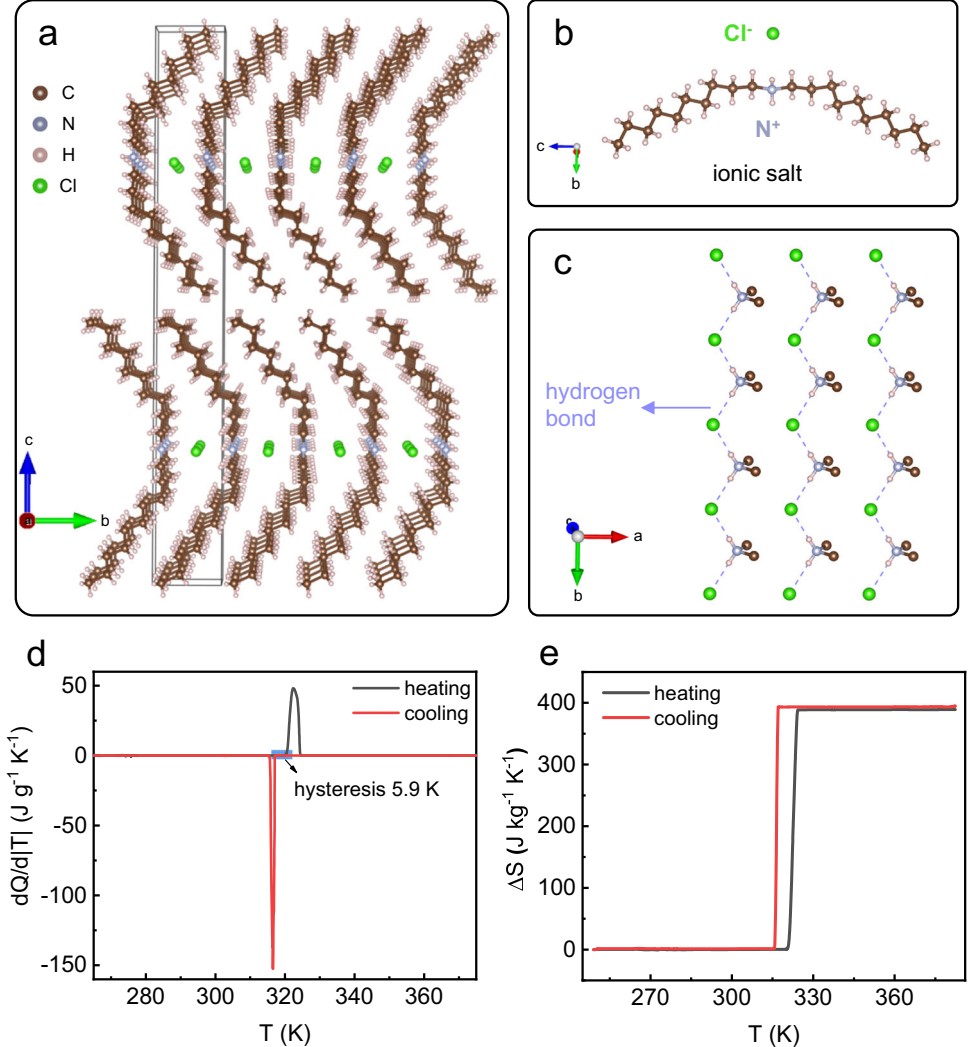

**Fig. 1 | Crystalline structure and thermal properties of phase transition for (CH₃–(CH₂)₉)₂NH₂Cl.** **a** Crystalline structure of $(CH_3–(CH_2)_9)_2NH_2Cl$ at 300 K, where the unit cell is marked by black frame. **b** Molecular structure of $(CH_3–(CH_2)_9)_2NH_2Cl$. **c** N–H...Cl hydrogen bond interaction in $(CH_3–(CH_2)_9)_2NH_2Cl$, where partial carbon and hydrogen atoms are omitted. **d** DSC measurement of $(CH_3–(CH_2)_9)_2NH_2Cl$ at atmosphere pressure with temperature ramping rate 0.1 K min⁻¹. **e** Temperature-dependence of entropy change across phase transition calculated from (**d**).

introducing cylindrical long chains in ammonium halide of trivial plastic crystal formed by globular species generates the novel crystal system[34–39]. We successfully synthesized $(CH_3–(CH_2)_{n-1})_2NH_2X$ ($n = 6$, 8, 10; X = Cl, Br) materials and single crystals for the first time. As our measurements show (Supplementary Table 1), solid-solid first-order phase transitions were affirmed by differential scanning calorimetry (DSC) at atmospheric pressure, intriguingly the thermal effect presented sizably and tunably. After cultivating high-quality single crystals of $(CH_3–(CH_2)_9)_2NH_2Cl$ (abbreviated as dC₁₀Cl) and $(CH_3–(CH_2)_9)_2NH_2Br$ (abbreviated as dC₁₀Br), SC-XRD under atmospheric pressure was performed and ordered crystalline state with emerged 2D layered structure was demonstrated at room temperature. Considering the remarkable thermal effect of phase transition and the transition temperature closer to room temperature, dC₁₀Cl is chosen for elucidation of 2D vdW structure and further exploration.

The crystal structure of dC₁₀Cl at room temperature is presented in Fig. 1a. The dC₁₀Cl molecules are packed with a triclinic crystal lattice, two molecules in each lattice (Supplementary Figs. S3, S4 and Table S2), and the lattice is parameterized with $a = 4.9$ Å, $b = 5.3$ Å, $c = 43.2$ Å, $\alpha = 89.6°$, $\beta = 89.7°$ and $\gamma = 88°$. As Fig. 1a shows, $((CH_3–(CH_2)_9)_2NH_2)^+$ alkylammonium cations and Cl⁻ anions, constituting the dC₁₀Cl molecules, are assigned orderly to form the layered structure, and the specific conformer of alkylammonium species is presented in Fig. 1b, where nitrogen atom (N⁺) of positive charge center is bonded with two hydrogen atoms and two long decyl chains $(CH_3–(CH_2)_9)$. Concretely, within each molecule layer of dC₁₀Cl, the nitrogen atoms (N⁺) locate at the center of $(CH_3–(CH_2)_9)_2NH_2^+$ and chlorine atoms (Cl⁻) are spread in *ab* plane, as shown in Fig. 1c, while $(CH_3–(CH_2)_9)$ chains, parallel to each other, are extended along the direction nearly perpendicular to the *ab* plane. Wherein, the N–H...Cl hydrogen bond interaction zigzag along the *b* axis (Fig. 1c), electrostatic interaction between N⁺ and Cl⁻, and lateral vdW force in the *ab* plane between the hydrocarbon chains are accountable for the intralayer interaction. Moreover, between the molecular layers, vertical vdW force along the *c* axis through the CH₃ groups at the end of $(CH_3–(CH_2)_9)$ chains maintain the structural stability (Fig. 1a). Therefore, the 2D layered structure is constructed by anisotropic interaction for intralayer and interlayer direction in dC₁₀Cl, dominantly strong intralayer ionic interaction and hydrogen bond interaction and weak interlayer vdW interaction.

The significant thermal effect of phase transition in dC₁₀Cl was further revealed by microcalorimetry performed on dC₁₀Cl single crystal at temperature changing rate of 0.1 K min⁻¹. As shown in Fig. 1d, a sharp first-order phase transition emerges at around $T_s \sim 320$ K, with

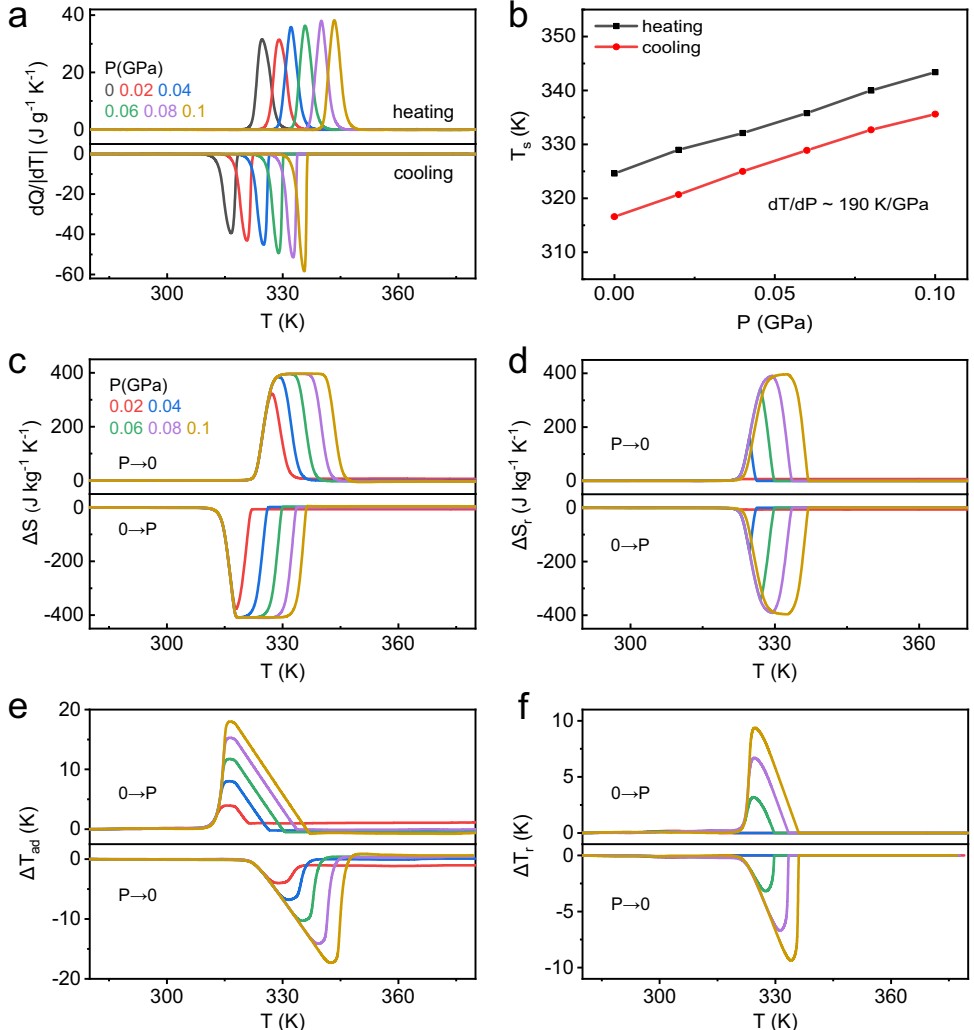

**Fig. 2 | Barocaloric performance of (CH$_3$−(CH$_2$)$_9$)$_2$NH$_2$Cl obtained by the quasi-direct method. a** Heat flow curves under variable pressure measured at a temperature rate 1 K/min. **b** Pressure-dependent phase transition temperature from (**a**). **c** Isothermal entropy change ΔS at variable pressure from the pressure-variable entropy curves (Supplementary Fig. S5). **d** Reversible entropy change ΔS$_r$ by overlapping of pressurization and depressurization from (**c**). **e** Adiabatic temperature change ΔT$_{ad}$ from pressure-variable entropy curves (Supplementary Fig. S7). **f** Reversible adiabatic temperature change ΔT$_r$ from heating entropy curve at atmosphere pressure and cooling entropy curves under applied pressure.

a low thermal hysteresis of 5.9 K determined as the distance between endothermic and exothermic peaks. The entropy change of phase transition (Fig. 1e), calculated by the integration of heat flow curves in Fig. 1d, attains ~400 J kg$^{-1}$ K$^{-1}$, which indicates the colossal caloric effect in 2D alkylammonium halide system analogous to that in 3D poly-alcohol plastic crystals[20,23]. Also, the thermal behavior of phase transition for dC$_{10}$Cl single crystal, i.e., transition temperature, entropy change and thermal hysteresis, is almost identical to that of polycrystal counterparts (Supplementary Table S1). The coexistence of large entropy change and low hysteresis in such 2D alkylammonium halide meets the prerequisite of colossal reversible BCE at low pressure, which is urgently attractive to be verified further.

**Barocaloric performances**

To investigate the barocaloric character of dC$_{10}$Cl, the pressure differential scanning calorimetry (P-DSC) was used to acquire the thermal behavior of phase transition under applied pressures and quasi-directly reveal the pressure-induced caloric effect. As shown in Fig. 2a, under stable applied pressure, the sample exhibits phase transition-related thermal response on heating and cooling with the rate of 1 K min$^{-1}$ (noted that the hysteresis can be larger for higher temperature scanning rates, which could critically influence the reversibility of the BCE

depending on the operation conditions; specific analysis can be seen in Supplementary Note 2). In the pressure range of 0–0.1 GPa, the transition temperature T$_s$ rises linearly with the applied pressure, exhibiting a magnificent rate of 190 K GPa$^{-1}$. The phase diagram of dC$_{10}$Cl in Fig. 2b exhibits the identically high sensitivity to pressure for endothermic and exothermic peaks, and consequently consistent thermal hysteresis of ~8 K under variable pressures. Also, identical entropy changes of phase transition under different pressures are revealed as ~400 J kg$^{-1}$ K$^{-1}$ (Supplementary Fig. S5). Including the entropy change from specific heat outside the phase transition region (Supplementary Fig. S6), the total entropy curves were constructed under variable pressures (Supplementary Fig. S7), on basis of which the pressure-induced isothermal entropy change (ΔS) and adiabatic temperature change (ΔT$_{ad}$) can be calculated quasi-directly. As shown in Fig. 2c, ΔS induced by pressure of 0.02 GPa reaches 332 J kg$^{-1}$ K$^{-1}$, and the maximum ΔS of ~400 J kg$^{-1}$ K$^{-1}$ can be realized by applying 0.06 GPa, which represents colossal barocaloric entropy change under low pressure. The colossal BCE is further evidenced by large ΔT$_{ad}$, as shown in Fig. 2e, ΔT$_{ad}$ as large as 18 K is induced by a pressure of 0.1 GPa.

Furthermore, to evaluate the barocaloric effect in thermodynamic cycle of practical significance, the reversible entropy change (ΔS$_r$) and reversible adiabatic temperature change (ΔT$_r$) were obtained quasi-

directly as Fig. 2d, f, respectively. The maximum $\Delta S_r$ of ~400 J kg$^{-1}$ K$^{-1}$ can be implemented under 0.08 GPa, which manifests that the complete phase transition of dC$_{10}$Cl can be reversibly driven by a low pressure of 0.08 GPa; for reversible $\Delta T_r$ induced by pressure (Fig. 2f), the maximum reaches 9.4 K at 0.1 GPa. Overall, due to the coexistence of large entropy change, high pressure-sensitivity, and low hysteresis of

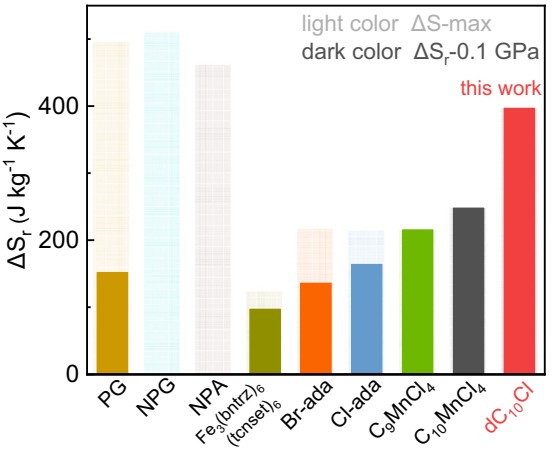

phase transition in dC$_{10}$Cl, the colossal reversible BCE under pressure as low as 0.1 GPa can be achieved. As shown in Fig. 3 and Table 1, the 2D dC$_{10}$Cl exhibits prominent phase transition with caloric effect as large as 3D plastic crystal polyalcohol (PG, NPG and NPA), but significantly the phase transition can be driven reversibly and completely by a low pressure of 0.1 GPa to induce the maximum entropy change. As a result, the reversible colossal caloric effect at 0.1 GPa exceeds that of all other reported barocaloric materials containing 3D plastic crystals, spin-crossover complex and hybrid organic–inorganic perovskites[20,21,23,25–29].

Moreover, the direct measurement (schematically in Fig. 4d) of reversible BCE was performed on dC$_{10}$Cl. The pressurization and depressurization processes are accomplished within 3 s, so the sample temperature was detected almost adiabatically. As shown in Fig. 4a, at ~330 K, the reversible adiabatic temperature change ($\Delta T_r$) of ~11 K can be induced by a pressure of 0.1 GPa, which is affirmed by cycle test of pressurization and depressurization (Fig. 4c). Noticeably, the ideal adiabatic cycle comprises the pressurization-induced temperature-increase ($T_0 \rightarrow T_1$) and depressurization-induced temperature-decrease ($T_1 \rightarrow T_0$). However, we failed to accurately measure the temperature change in the depressurization process since the applied pressure cannot be fully maintained but spontaneously release some due to our device. Concretely in Fig. 4a and contrastive illustration in Fig. 4b, the temperature change in the first process (I) reflects the thermal effect induced by approximately adiabatically pressurization to 0.1 GPa, and in the second process (II) involving the spontaneous release of pressure, the heat exchange between the heated sample and environment with temperature of ~330 K contributes to the cooling trend; additionally in the third process (III), at initial temperature of ~330 K, $\Delta T_{ad}$~6 K can be induced by depressurization where the $\Delta T_{ad}$ is underestimated due to the spontaneous release of pressure in the second process (II), and similar to the second process (II), heat

**Fig. 3 | The comparison of reversible barocaloric entropy change of our dC$_{10}$Cl with other reported colossal barocaloric materials.** The materials include 3D plastic crystals PG, NPG, NPA, spin-crossover Fe$_3$(bntrz)$_6$(tcnset)$_6$, substituted adamantane Br-adamantane, Cl-adamantane and hybrid organic–inorganic perovskite (C$_n$H$_{2n+1}$NH$_3$)$_2$MnCl$_4$ ($n$ = 9, 10)[20,21,23,25–29]. Light color denotes the entropy change $\Delta S$ of phase transition, while dark color marks the reversible entropy change $\Delta S_r$ driven by a hydrostatic pressure of 0.1 GPa.

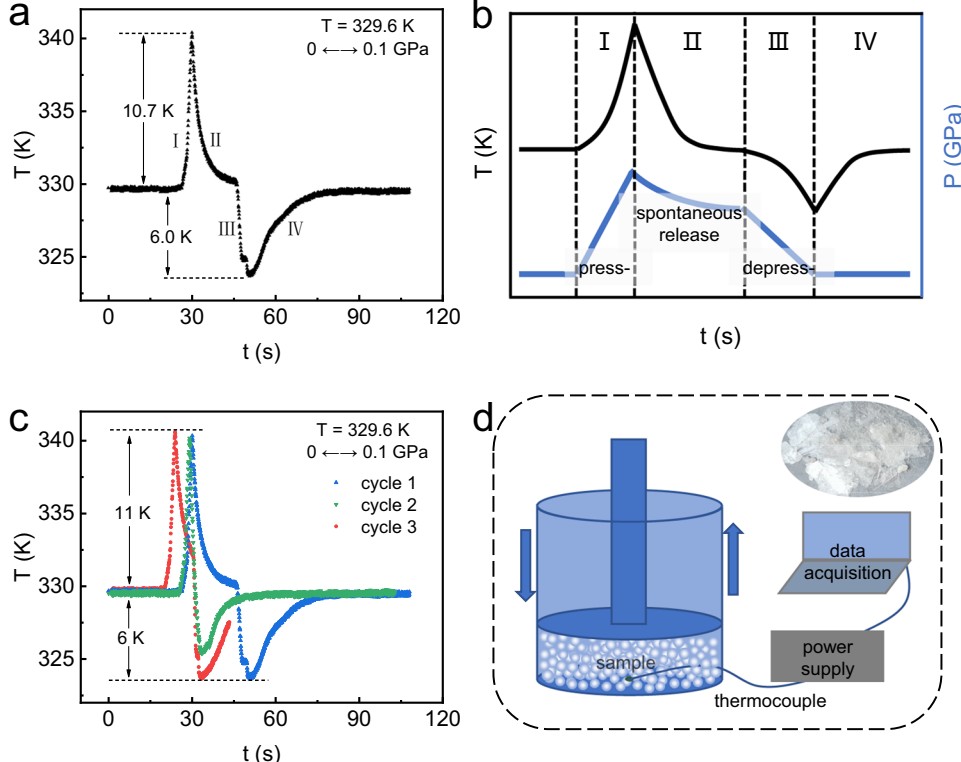

**Fig. 4 | The direct measurement of adiabatic temperature change $\Delta T_{ad}$ driven by pressure in dC$_{10}$Cl. a** $\Delta T_{ad}$ induced by pressure of 0.1 GPa. **b** Schematic illustration of temperature change and pressure evolution during the adiabatic test, containing the processes of pressurization (press-) and depressurization (depress-). **c** Cycle measurements of $\Delta T_{ad}$ induced by pressure of 0.1 GPa. **d** Schematic diagram of direct measurement of $\Delta T_{ad}$, where the sample's image is for indication only, and the real image of single crystal sample is shown at top right.

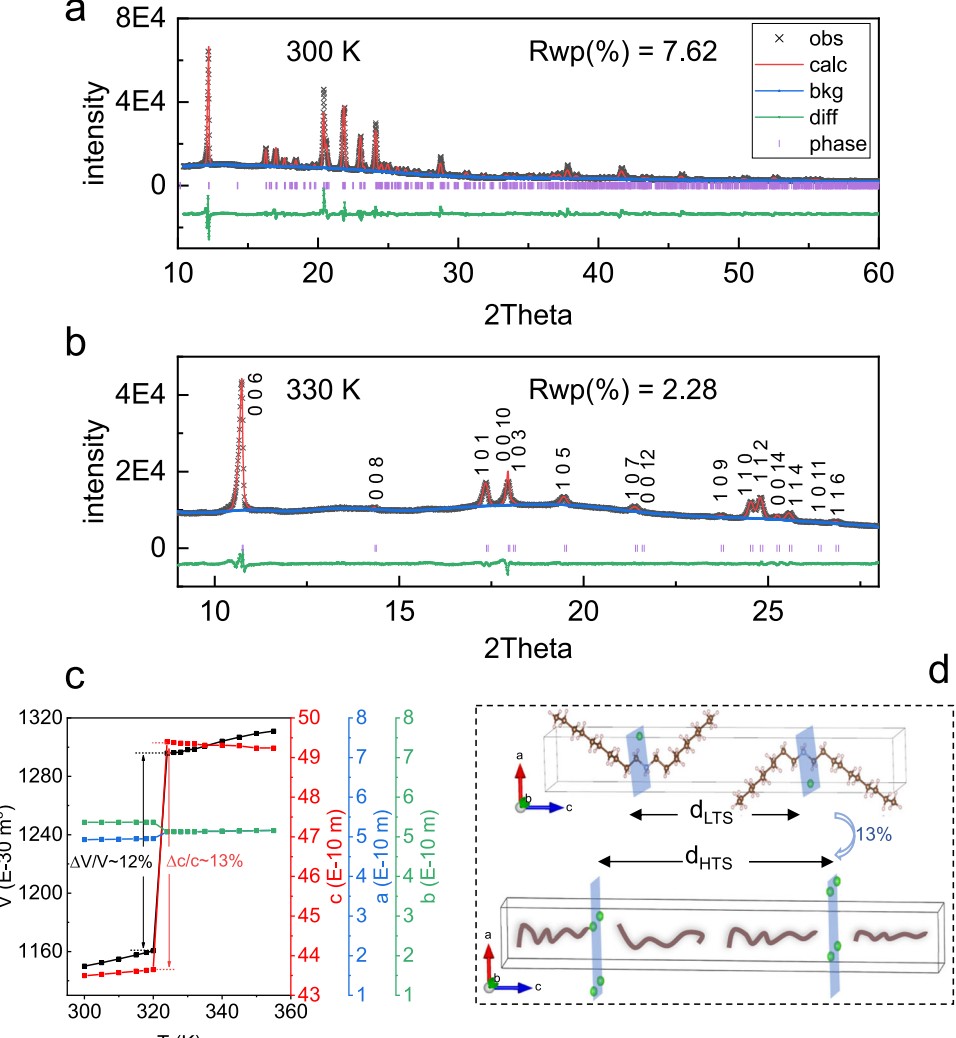

**Fig. 5 | Powder x-ray diffraction result of the $(CH_3-(CH_2)_9)_2NH_2Cl$. a** Refinement result of diffraction pattern at 300 K for low-temperature-state in $(CH_3-(CH_2)_9)_2NH_2Cl$. **b** Le Bail method fitting result of diffraction pattern at 330 K for high-temperature-state in $(CH_3-(CH_2)_9)_2NH_2Cl$, where the observed (black), calculated patterns (red), their difference (green), peak positions (purple bar), background (blue), and error factor Rwp are provided. **c** The evolution of lattice parameters and unit cell volume with temperature in $(CH_3-(CH_2)_9)_2NH_2Cl$. **d** Schematic illustration of layer spacing expansion across the phase transition in $(CH_3-(CH_2)_9)_2NH_2Cl$.

exchange between the sample and environment causes temperature rising back during the fourth process (IV). However, the good repeatability on pressurization during cycle test demonstrates the reliability of measured $\Delta T_r$ (Fig. 4c).

Moreover, larger $\Delta T_r$ of ~22 K can be achieved under pressure of 0.2 GPa at 330 K (Supplementary Fig. S9), but further enlarging applied pressure as 0.3 GPa cannot enhance the $\Delta T_r$ any more (Supplementary Fig. S10), indicating the maximum of adiabatic temperature change in $dC_{10}Cl$ is ~22 K. The superior $\Delta T_r$ at low pressure can be revealed by comparing the $dC_{10}Cl$ with all the reported materials measured by direct measurement method (Supplementary Fig. S11). Hitherto the colossal reversible barocaloric effect of 2D alkylammonium halide $dC_{10}Cl$ has been evidenced by both quasi-direct and direct measurements, and the underlying mechanism of intriguing phase transition will be discussed further.

**Structural transition**

From SC-XRD, although the structure was detailed at 300 K below transition temperature $T_s \sim 320$ K for $dC_{10}Cl$, we were unable to obtain regular diffraction information via SC-XRD at above $T_s$ though a partial stacking order remains (Supplementary Fig. S2). To disclose the

structural evolution of 2D vdW alkylammonium halide $dC_{10}Cl$ across $T_s$, the temperature-variable PXRD under atmospheric pressure was performed on $dC_{10}Cl$ at 300–355 K. And the resulted diffraction patterns (Supplementary Fig. S12) show that $dC_{10}Cl$ exhibits a transition onset of lattice symmetry at 322 K, then the phase transition gets accomplished at 324 K, which is in perfect accordance with the results from DSC characterization above. Further Rietveld refinement of diffraction patterns (Supplementary Figs. S13–17) demonstrates that at low-temperature-state (LTS) below 322 K, $dC_{10}Cl$ maintains the triclinic lattice with space group of P-1, specifically $a = 4.9$ Å, $b = 5.4$ Å, $c = 43.5$ Å, $\alpha = 90°$, $\beta = 89.8°$ and $\gamma = 88.2°$ at 300 K ($a = 4.9$ Å, $b = 5.3$ Å, $c = 43.2$ Å, $\alpha = 89.6°$, $\beta = 89.7°$ and $\gamma = 88°$ from SC-XRD, 300 K), where the carbon chains present definite conformer (Fig. 5a). Such refined molecular configuration of $dC_{10}Cl$ at 300 K from PXRD fits well with the result resolved from SC-XRD patterns (Supplementary Table S6).

For the high-temperature-state (HTS) higher than 324 K, although the organic chains become disordering and lose translational symmetry with definite atomic position, the crystalline structure composed of inorganic Cl⁻ ions can be confirmed by Le Bail method (Supplementary Figs. S18–26, Fig. 5d), where reliable lattice parameters can be obtained before detailed atomic position fitting[40]. As

shown in Fig. 5b, the lattice at HTS, at 330 K for instance, is body-centered tetragonal, presenting space group of I4/mmm for instance, according to the observed reflection peaks with the even sum of $h$, $k$, and $l$ induced by extinction rule of body-centered space group. Specifically, the tetragonal lattice at 330 K is parameterized with $a = b = 5.13$ Å and $c = 49.35$ Å. It should be noted that the carbon chains would exhibit multiple variable conformations, while the $Cl^-$ anions become less bonded by the organic chains (see IR spectra below). Hence it is understandable that the crystalline structure composed of inorganic $Cl^-$ ions adapts the higher symmetry at HTS time-averaged and space-averagely, wherein the $Cl^-$ anions can be easily accommodated in the high symmetry lattice (schematically in Fig. 5d).

From the refined lattice information, it is revealed that the interlayer expansion along $c$ axis can be as large as 13% across $T_s$ (as shown in Fig. 5c and illustrative transition from $d_{LTS}$ to $d_{HTS}$ in Fig. 5d), while $ab$ plane remains nearly unchanged (4.9 Å × 5.3 Å at LTS (300 K), 5.13 Å × 5.13 Å at HTS (330 K)), consequently volume expansion as much as 12% ($\Delta V$ ~ 11.9 E-5 $m^3$ $kg^{-1}$) occurs during phase transition (Fig. 5c). The specific anisotropic expansion can be ascribed to the layered 2D structural character in $dC_{10}Cl$; within the 2D structure, the weak interlayer vdW interaction along $c$ axis dominantly contributes to the prominent expansion across the phase transition.

From the underlying relationship between the volume change and thermal effect of phase transition, such an extraordinary large expansion along the length direction of carbon chains (Fig. 5d) can endow sufficient free volume for carbon chains between N–Cl planes, potentially allowing for great conformational disorder of carbon chains and consequently large entropy change in $dC_{10}Cl$. And the interlayer spacing enlargement-induced volume change of phase transition in $dC_{10}Cl$ is shown to be largest among the colossal barocaloric materials (Table 1). Based on the Clausius-Clapeyron relation ($dT_s/dP = \Delta V/\Delta S$), the large entropy change across the phase transition tends to induce the low pressure-sensitivity of phase transition. Significantly, in the 2D vdW $dC_{10}Cl$, the volume change of phase transition appears to be colossal, due to the weak interlayer vdW interaction. In this case, the large pressure sensitivity of phase transition with colossal entropy change can be accomplished with the compensation of the large volume expansion. Therefore, the coexistence of colossal entropy change and magnificent volume change during the phase transition contributes to the superior BCE reversibly driven by low pressure in 2D vdW alkylammonium halide system $dC_{10}Cl$. The conformational dynamics of carbon chains during phase transition in $dC_{10}Cl$ is intriguing to be explored, for elaborating the mechanism of peculiar phase transition therein.

## Molecular dynamics simulation

Molecular dynamics (MD) simulation was performed on $dC_{10}Cl$ to clarify the conformational evolution of organic chains, which is generally difficult to be detected experimentally for the intricacy of organic chain conformations at HTS. Based on the molecular structure parameters from SC-XRD at 300 K, we constructed supercell and performed MD simulations. See details in Methods.

As shown in Supplementary Fig. S27, a first-order phase transition was deduced by MD simulation at ~430 K in $dC_{10}Cl$, where the overestimated transition temperature can be attributed to the superheating problem in MD simulations, i.e., the perfect crystal without surfaces and defects in simulation contrary to the real materials where phase transition onset could be at surfaces and defects[41]. Concretely at simulated LTS (Fig. 6a), the molecules stay aligned regularly and organic chains keep uniform conformation. Contrastingly at simulated HTS (Fig. 6b), the conformation of organic chains become disordered prominently, also the $Cl^-$ anions bear positional disorder arising from the intensified vibration.

The conformational transition of organic chains can be illustrated quantitatively by the statistical result of radial distribution function (Fig. 6c, d) and dihedral angle distribution (Fig. 7). Radial distribution function $g(r)$ refers to the variation of atomic density as a function of distance from a reference atom (as schematically shown in Fig. 6e); it is an indication of atomic packing pattern[42]. The temperature-variable $g(r)$ of carbon atoms in $dC_{10}Cl$ are shown in Fig. 6c, the trend of $g(r)$ at simulated LTS (300 K and 400 K) differing from that at simulated HTS (500 K and 600 K). For the atomic distance of 0–5 Å, $g(r)$ presents multiple sharp peaks at LTS, where each peak ($p_1$–$p_4$ in Fig. 6c) refers to each radial distance between carbon atoms in carbon chains with short-range ordered trans conformer (Fig. 6e). Across the LTS–HTS transition, the $p_4$ peak in $g(r)$ tends to disappear, indicating the conformal disordering of C–C–C–C–C group with 4 C–C bonds, while partially ordered C–C–C–C with 3 C–C bonds still exists at HTS. For the range of 5–20 Å (Fig. 6f), $g(r)$ presents multiple peaks at LTS while getting flat after the phase transition, which manifests the carbon chains transform from the long-range ordered conformer chains aligned regularly to the disordered conformer chains without regular arrangement[43]; Fig. 6f schematically represents that the conformational transition of carbon chains during phase transition gives rise to the distribution disorder of interchain and intrachain carbon atoms, consequently rendering the radial distance getting uniform at HTS.

More explicit demonstration of carbon chain conformer evolution during the phase transition can be given by temperature-variable C–C–C–C dihedral angle composed of 3 C–C bonds (Fig. 7). The C–C–C–C dihedral angle refers to the angle between the plane formed by the first two C–C bonds and the plane formed by the last two C–C bonds, thus indicative of the local conformer of C–C–C–C group. Specifically, the trans conformer exhibits the dihedral angle of around 180°, while the rotation of partial chain around the carbon bonds induces the appearance of various chain conformation and dihedral angles, as schematically shown in Fig. 7b. In Fig. 7a, the C–C–C–C dihedral distributions at different temperatures reveal the distinguishable motifs of C–C–C–C dihedral angles before and after the phase transition. At LTS, the bands concentrate around 160°–180° and 60°–90°, where the former denotes trans-like conformer and the latter denotes the gauche conformer in carbon chains. In accordance with the SC-XRD result, the N–H...Cl hydrogen bond interaction between the $((CH_3–(CH_2)_9)_2NH_2)^+$ group and $Cl^-$ anions induces the gauche conformer of C–C–C–C closest to the $Cl^-$ framework, and others are still in trans (see Fig. 1b). At HTS, more C–C–C–C dihedral angles are concentrated around 60°–90°, while there are less for 160°–180°; except the two categories, other angles emerge at HTS, which denotes the transient state during the dynamic trans-gauche conformer transition. The dispersive C–C–C–C dihedral angles at the HTS manifests the multiplying conformers coexisted in carbon chains, which refers to the specific disorder of organic chains in $dC_{10}Cl$.

The conformation disordering of organic chains related to the free twisting at carbon chains gives rise to the colossal entropy change during the order–disorder phase transition[27–29,44], which accounts for the large entropy change in $dC_{10}Cl$. Moreover, the enthalpy change of phase transition per mole for $dC_{10}Cl$ approaches double that for n-decane (Supplementary Table S7), which indicates the disordering of organic chains is analogous to the incomplete melting-like free twisting and conformation disorder of organic chains.

## Molecular infrared vibration

To investigate the interaction evolution at the level of atomic group during the phase transition, the molecular infrared vibration behaviors before and after the phase transition were explored (Fig. 8). We measured the temperature-variable infrared spectra in the wavenumber range of 900–4000 $cm^{-1}$ from 295 K to 353 K on heating. The change of

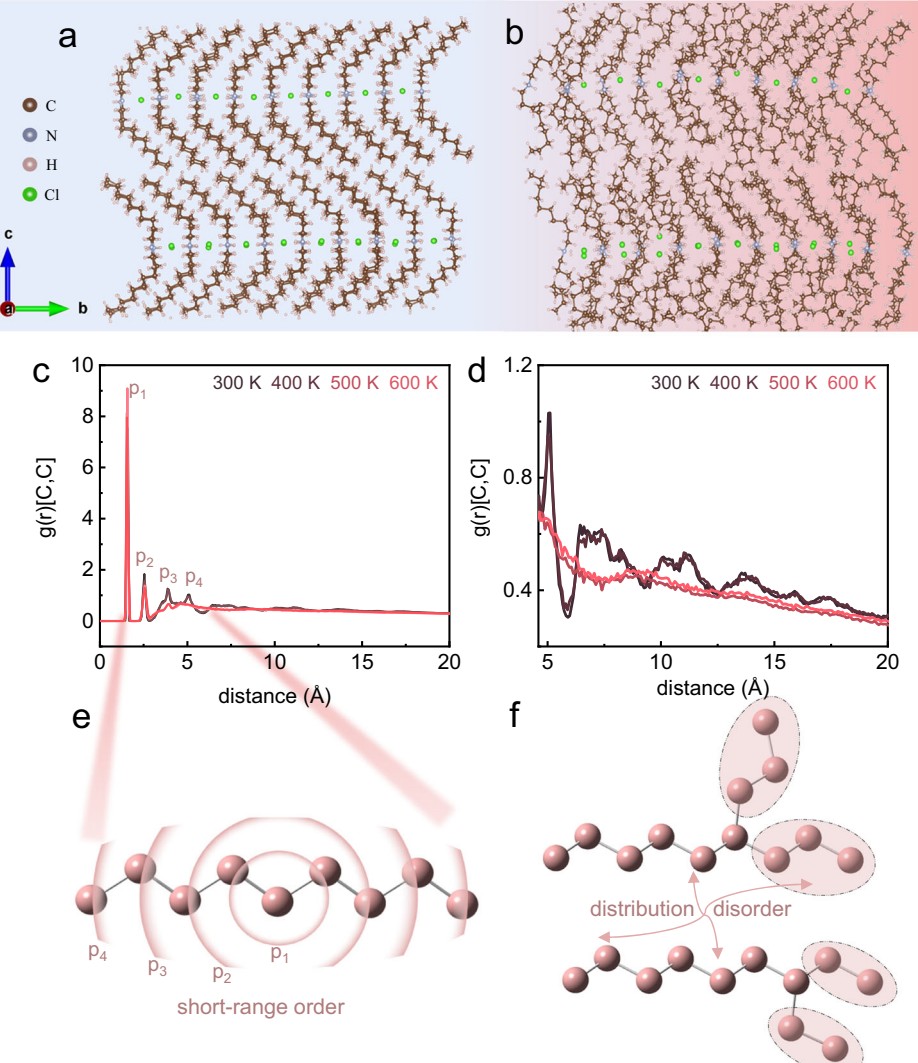

**Fig. 6 | MD simulation on the (CH₃–(CH₂)₉)₂NH₂Cl.** MD results of molecular crystal structure are shown in (**a**) for 300 K and (**b**) for 500 K. **c** The radial distribution function of carbon atoms at variable temperatures in the range of 0–20 Å.

**d** Enlarged zone of 5–20 Å from (**c**). **e** The schematic demonstration of short-range order of carbon chain. **f** The schematic demonstration of distribution disorder of carbon chain at long-range level.

molecular vibration modes occurs across the LTS–HTS phase transition at 323–326 K, which is in accordance with the DSC and XRD results. The vibration modes on $NH_2$, $CH_2$ and $CH_3$ groups along the entire organic chains (Fig. 8a) are elaborated below.

For the $NH_2$ group at the center of $(CH_3–(CH_2)_9)_2NH_2^+$ (Fig. 8a), the N–H symmetric and asymmetric bending vibrations appear in the range of 1500–1600 cm⁻¹ [45,46]. As shown in Fig. 8b, at LTS, the $NH_2^+$ bending mode is split at three frequencies of 1593 cm⁻¹, 1559 cm⁻¹ and 1541 cm⁻¹, which is due to the different length of N–H...Cl hydrogen bonds (consistent with the SC-XRD result in Supplementary Table S5). Across the LTS–HTS transition, the intensity of bending vibrations with higher frequency at 1593 cm⁻¹ enhances meanwhile the vibrations at 1541 cm⁻¹ and 1559 cm⁻¹ tend to disappear, which indicates the weakening of N–H...Cl hydrogen bonds and nonequivalent-equivalent environment change of hydrogen bonds during LTS–HTS phase transition. Moreover, as shown in Fig. 8e, N–H stretching modes appear at 2787 cm⁻¹ and 2984 cm⁻¹ at LTS [47], which get weakened across $T_s$, demonstrating the change of hydrogen bond environment during phase transition as well [47]. In addition, at LTS, the dispersive broad absorption bands in the range of 2300–2600 cm⁻¹ (Fig. 8d) can be assigned as $NH_2^+$ stretching vibrations with N–H...Cl hydrogen bond environment [48]. The dynamic motion of $NH_2^+$ group result in the

continuously variable hydrogen bond interaction, which induces the variable N–H distance and consequently variable N–H stretching vibration frequencies; hence multiple bands appear successively at 2300–2600 cm⁻¹ at LTS. At HTS, the intensity of dispersive $NH_2^+$ stretching bands gets decreased, indicating that the effect of hydrogen bond interaction on $NH_2^+$ group gets weakened across the phase transition.

As for the $CH_2$ group at the body of carbon chains (Fig. 8a), two bands at 1467 cm⁻¹ and 1482 cm⁻¹ are assigned to the C–H scissoring vibrations of $CH_2$ group at the LTS (Fig. 8b) [46], where the band splitting of C–H scissoring vibrations indicates the interchain vibrational coupling [46]. With the LTS-HTS transition, the two bands suddenly merge into one with lower frequency (Fig. 8c), which indicates the interaction of intralayer chains get weakened [49].

Moreover, for the $CH_3$ group at the tail of carbon chains (Fig. 8a), C–H symmetric and asymmetric stretching vibration bands are assigned at 2869 cm⁻¹ and 2953 cm⁻¹ (Fig. 8e) [46], and their blueshift across $T_s$ on heating indicates that the $CH_3$ group get less restricted and underlyingly vdW interaction between the carbon chains along their extension direction (c axis) becomes weak on heating; that is, the interlayer vdW interaction is weakened; also, the $CH_3$ umbrella deformation mode appears at 1376 cm⁻¹ at LTS,

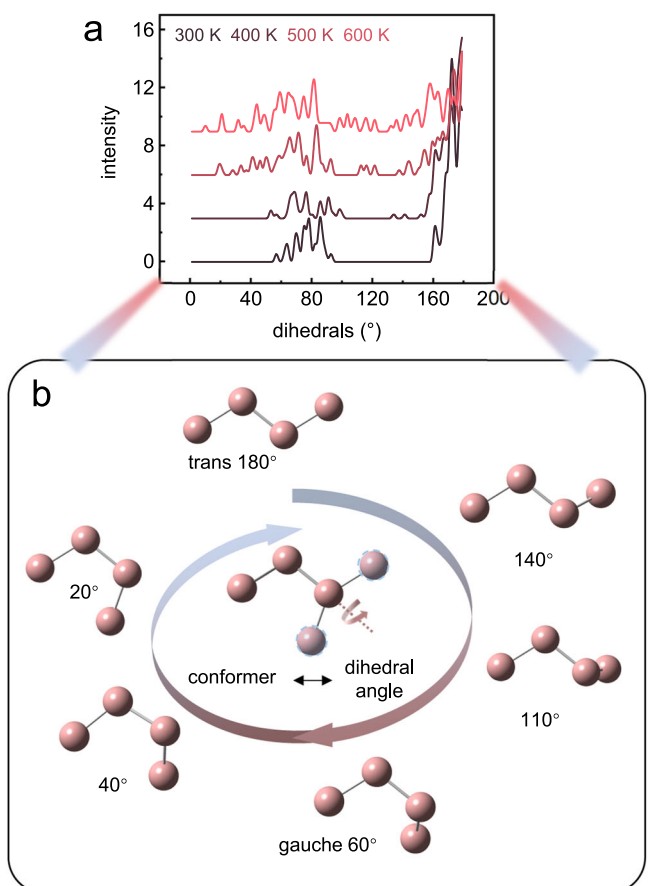

**Fig. 7 | The distribution of C–C–C–C dihedral angle composed of 3 C-C bonds from the MD simulation for (CH₃–(CH₂)₉)₂NH₂Cl. a** The dihedral angle distribution at variable temperatures. **b** The schematic relation between the carbon chain conformer and the dihedral angle.

shifting to 1379 cm⁻¹ at HTS (Fig. 8b, c) due to the effect of interlayer interaction weakening[46]. All these are in accordance with the effect caused by the extraordinary enlargement of interlayer spacing along *c* axis (Fig. 5d).

Overall, undergoing the LTS-HTS transition, the intralayer N−H··· Cl hydrogen bond interaction gets weakened, which renders the freer motion of ammonium polar head (NH₂⁺), also the intralayer and interlayer vdW interaction between carbon chains gets weakened across the phase transition, accounting for the disordered conformations of organic chains with less hindrance at HTS and the colossal entropy change of phase transition. Furthermore, the significantly weakening interlayer vdW interaction through CH₃ group specifically gives rise to the extraordinary expansion of interlayer spacing along *c* axis, critically leading to the large volume change of phase transition in such 2D alkylammonium halide, which accords well with the PXRD results shown in Fig. 5.

## Discussion

Conventional 3D plastic crystals enable intensive orientational disorder of molecules at plastic crystalline state, which is favored by the high symmetry of relatively small molecule and cubic lattice[50]. Although producing colossal entropy change from molecular orientational order-disorder transition, the 3D plastic crystals exhibit relatively limited volume change across the phase transition and considerable incompatibility of ordered crystalline and plastic crystalline structures, which generally induces large hysteresis in most 3D plastic crystals[23] (Table 1). Therefore, the characteristics of large

entropy change, large hysteresis and relatively limited volume expansion contribute to low pressure-sensitivity and large pressure hysteresis of phase transition, hindering the colossal caloric effect reversibly driven by low pressure in 3D plastic crystals. After introducing the lonog alkyl chains in 3D plastic crystals, as for the resulted 2D vdW dialkylammonium halide (CH₃−(CH₂)₉)₂NH₂Cl in this work, the alignment of long carbon chains constructs the anisotropic interaction in crystal, involving strong intralayer hydrogen bond perpendicular to the direction of chain extension and weak interlayer vdW force along the direction of chain extension. Leveraging the 2D character, there forms the phase transition during which interlayer vdW interactions get significantly weakened and interlayer spacing significantly enlarges, hence drastic orientational disorder of partial group emerges in long ((CH₃−(CH₂)₉)₂NH₂)⁺ chains, i.e., conformational disorder of organic chains; consequently, the colossal entropy change and large volume change of phase transition can be realized altogether. Also, the interlayer weak vdW force-related 2D structural feature is deduced to induce low energy barrier of phase transition and consequently low hysteresis, for instance the hysteresis lower than 10 K in present dialkylammonium halide system and the hysteresis lower than 5 K in hybrid organic−inorganic layered perovskites[28,29]. Therefore, the combination of large entropy change, large volume change-related high pressure-sensitivity and weak vdW force-related low hysteresis of phase transition contributes to the enhanced reversibility of BCE meanwhile maintaining the advantage of colossal thermal effect in 3D plastic crystals, accomplishing the colossal reversible BCE driven by low pressure in 2D vdW alkylammonium halides.

To conclude, leveraging quasi-direct and direct barocaloric measurement, we report the reversible colossal barocaloric effect in the 2D vdW dialkylammonium halides (CH₃−(CH₂)ₙ₋₁)₂NH₂X (X: halogen element). The reversible entropy change attains to $\Delta S_r \sim 400 \, \text{J kg}^{-1} \text{K}^{-1}$ while the directly measured adiabatic temperature change reaches $\Delta T_{ad} \sim 11 \, \text{K}$ under a pressure lower than 0.1 GPa in the representative (CH₃−(CH₂)₉)₂NH₂Cl, surpassing all other reported barocaloric materials. Combining SC-XRD, PXRD, MD simulations with IR spectra analysis, the colossal thermal effect is indicated to stem from a phase transition with dramatic order-disorder conformation change of organic chains within the 2D layered structure. Across the phase transition, the prominent weakening of interlayer vdW interactions along extension direction of carbon chains (c axis) induces the significantly enlarged volume ($\Delta V/V \sim 12\%$; $\Delta c/c \sim 13\%$) for the accommodation of organic chains with sufficiently disordered conformation, rendering the large volume change, large entropy change and related low hysteresis of phase transition; ultimately, the colossal thermal effect can be driven reversibly by low pressure, generating the reversible colossal barocaloric effect. This work presents an emergent barocaloric mechanism available for the design strategy of barocaloric cooling refrigerant by constructing 2D vdW alkylammonium halides with long carbon chains cooperated.

## Methods
### Sample preparations

(CH₃−(CH₂)ₙ₋₁)₂NH₂X (X = Cl, Br) materials were synthesized by a reaction in anhydrous ethyl alcohol. The preparation process below produces polycrystal samples. Generally, the dialkylamine ((CₙH₂ₙ₊₁)₂NH) and hydrochloric acid (mass fraction of 0.37) were weighed at a molar ratio of 1:1, and then added into the ethyl alcohol, the hydrochloric acid being added drop by drop. The mixture was heated and stirred under reflux for six hours, then on cooling, precipitate appears slowly. For purification, the product was recrystallized at least for three times with anhydrous ethyl alcohol.

Specifically, for cultivating single crystal of (CH₃−(CH₂)ₙ₋₁)₂NH₂X (n = 10; X = Cl, Br) samples, recrystallization were conducted via

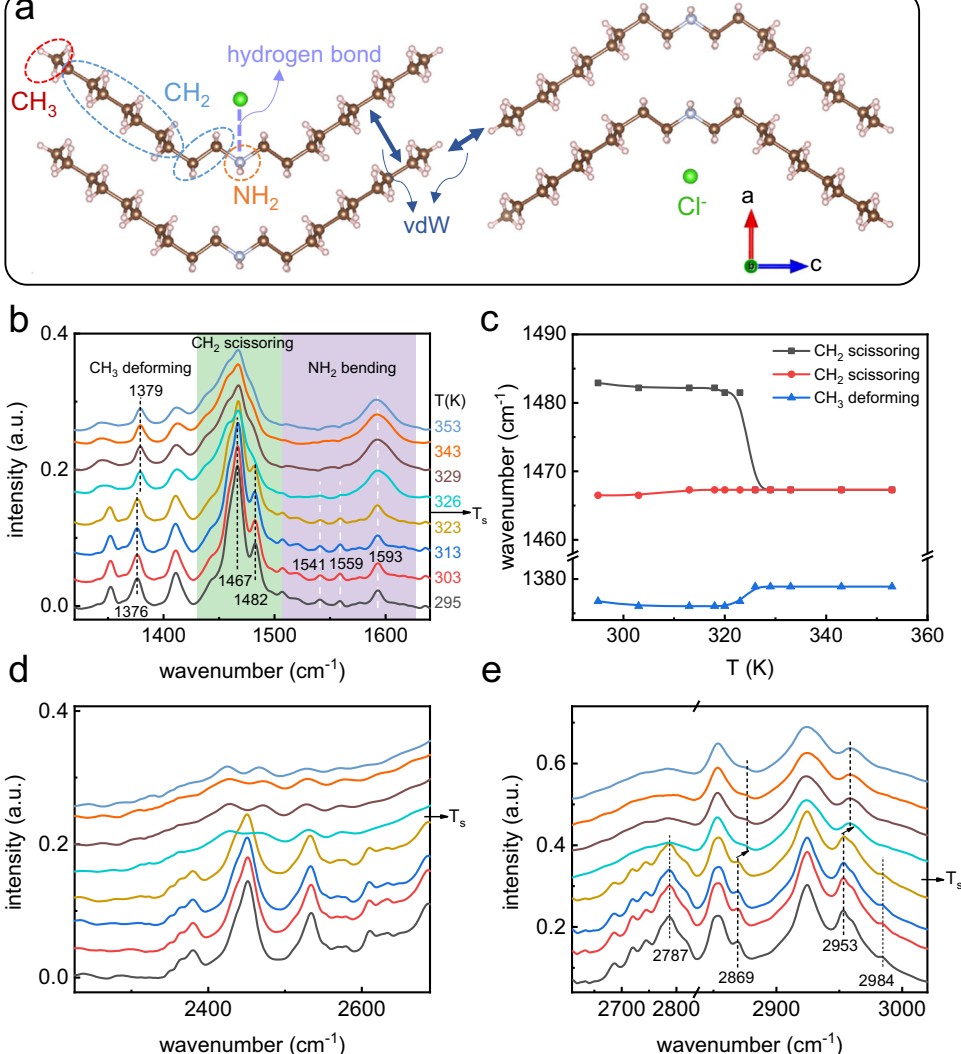

**Fig. 8 | Temperature-variable infrared spectra of (CH₃−(CH₂)₉)₂NH₂Cl. a** The interchain and interlayer vdW interactions and the hydrogen bond interaction of organic chains with Cl⁻ anions in $(CH_3-(CH_2)_9)_2NH_2Cl$. **b** Temperature-variable infrared spectra in the range of 1300–1650 cm⁻¹. **c** Temperature-dependent vibration frequency of specific vibration band in (**b**). **d, e** Temperature-variable infrared spectra in the range of 2300–2700 cm⁻¹ and 2700–3000 cm⁻¹, respectively.

slow evaporation for one week without any motion and interference. Then high-quality single crystal of $(CH_3-(CH_2)_9)_2NH_2Cl$ and $(CH_3-(CH_2)_9)_2NH_2Br$ could be obtained.

### High resolution single crystal x-ray diffraction (SC-XRD)

at variable temperatures was performed by BRUKER D8 VENTURE with $\lambda = 0.71073$ Å. The datasets of $(CH_3-(CH_2)_9)_2NH_2Cl$ were collected on heating using the Multi-Scan method (SADABS). The structure was solved and refined using the Bruker SHELXTL Software Package.

### Powder x-ray diffraction (PXRD)

at variable temperatures was measured by Rigaku Smartlab using Cu Kα radiation. The structure of low-temperature-state was refined by Rietveld method using the General Structure Analysis System (GSAS) suite, while the lattice parameters of high-temperature-state was obtained by Le bail method using the GSAS suite.

### Pressure differential scanning calorimetry (P-DSC)

was used to characterize barocaloric effect by measuring heat flow under pressure. The pressure μDSC7 evo microcalorimeter (SETARAM,

France) was employed, which provides pressure 0–0.1 GPa by compressed N₂ gas with high purity (99.999%).

### Infrared (IR) spectra

were measured using a BRUKER TENSOR II FTIR (Fourier Transform Infrared) spectrometer in the frequency range of 900–4000 cm⁻¹ with 1.5 cm⁻¹ resolution.

### The classic molecular dynamics (MD) simulations

were performed using the LAMMPS package with periodic boundary condition[51]. The reactive force field (ReaxFF) was utilized in our MD simulations[52]. The Newton equation of motion was integrated by the velocity Verlet algorithm and the time-step is 0.5 fs. Based on the lattice parameters from SC-XRD at 300 K for $(CH_3-(CH_2)_9)_2NH_2Cl$, we constructed a 8*8*2 supercell at low-temperature-state. MD simulations were performed by heating the supercell from 1 K to 800 K in the NPT ensemble (3,000,000 time steps), then tracking the supercell structure at variable temperature for low-temperature-state and high-temperature-state. The radial distribution function and dihedral angle distribution are obtained via I.S.A.A.C.S. (Interactive Structure Analysis of Amorphous and Crystalline Systems) on the MD results.

**Direct measurement of adiabatic temperature change $\Delta T_{ad}$ driven by pressure**

The quasi-adiabatic condition was constructed by an isothermal bath combined with the fast operation of pressurization and depressurization. At constant temperatures set by isothermal bath, the sample containing K-type thermocouple was pressurized up to set point within 3 s, then $\Delta T_{ad}$ induced by pressurization can be measured; however the target pressure cannot be fully maintained but spontaneously release some due to our device, then the sample decreased to bath temperature for the thermal equilibrium with environment; after sample kept at bath temperature, the remained pressure was released within 3 s, then the underestimated $\Delta T_{ad}$ induced by depressurization can be obtained. Through several pressurization-depressurization cycles, the reversible adiabatic temperature change ($\Delta T_r$) at bath temperature can be obtained. The schematic diagram for directly measuring $\Delta T_{ad}$ is shown in Fig. 4d.

**Calculation of barocaloric entropy change and adiabatic temperature change**

Based on the heat flow (Q) response of temperature at variable pressure, the phase transition entropy curves can be constructed via the integration of heat flow[53]:

$$S_{pt}(T,P) = \int_{T_0}^{T} \frac{1}{T'} \frac{Q(T',P)}{\frac{dT'}{dt}} dT' \tag{1}$$

Considering the thermal response besides the latent heat of phase transition, the total entropy curves can be constructed including the specific heat capacity contribution, where specific heat capacity $C_P$ at variable pressures were approximated as that for atmosphere pressure:

$$S(T,P) = S_{pt}(T,P) + \int_{T_0}^{T} \frac{C_p}{T'} dT' \tag{2}$$

Isothermal entropy change curves ($\Delta S_P$) can be obtained quasi-directly by isothermal subtraction of entropy curves at variable pressure: $\Delta S_P(T, P_0 \rightarrow P_1) = S(T,P_1) - S(T,P_0)$. Concretely, the $\Delta S_P$ for pressurization process ($P_0 < P_1$) utilizes the Q(T, P) on cooling, while the $\Delta S_P$ for depressurization ($P_0 > P_1$) process utilizes the Q(T, P) on heating. And reversible isothermal entropy change $\Delta S_r$ can be treated as the overlapping of $\Delta S_P$ between pressurization and depressurization processes.

Adiabatic temperature change curves ($\Delta T_P$) can be obtained quasi-directly by adiabatic subtraction of entropy curves at variable pressure: $\Delta T_p(T, P_0 \rightarrow P_1) = T(S,P_1) - T(S,P_0)$. Similar to $\Delta S_P$, the $\Delta T_P$ for pressurization process should be decided by entropy curves on cooling, while the $\Delta T_P$ for depressurization process be decided by entropy curves on heating. Reversible adiabatic temperature change $\Delta T_r$ can be obtained by the adiabatic subtraction of heating entropy curve at atmosphere pressure and cooling entropy curve at applied pressure[54].

## Data availability

The main data supporting the findings of this study are available within the paper and its Supplementary Information. Considering the huge quantity of raw data, all raw data generated during the current study are available from the corresponding author (F.X.H.) upon request.

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

## Acknowledgements

This work was supported by the Science Center of the National Science Foundation of China (52088101(B.-G.S.)), the National Key Research and Development Program of China (2021YFB3501202 (F.-X.H.), 2020YFA0711500 (J.W.), 2019YFA0704900 (F.-X.H.), 2022YFB3505201 (J.W.), 2021YFA1400300 (B.-G.S.), 2023YFA1406003 (F.-X.H.)), the National Natural Sciences Foundation of China (92263202 (F.-X.H.), U23A20550 (F.-X.H.), 22361132534 (J.W.), 51971240 (J.W.)), the Strategic Priority Research Program B (XDB33030200 (B.-G.S.)) of Chinese Academy of Sciences (CAS), and the Synergetic Extreme Condition User Facility (SECUF). The authors acknowledge Dr. Yan Chen (North China Electric Power University, Beijing 102206, China) for her help during the RDG calculation.

## Author contributions

F.X.H. and B.G.S. formulated the project. Y.H.G. synthesized the compounds. Y.H.G., Q.Z.H., Z.Y.T. and B.J.W collected and analyzed the powder x-ray diffraction data. Y.H.G. and Y.T.S. collected and analyzed single-crystal x-ray diffraction data. Y.H.G. collected and analyzed pressure calorimetry data. Y.H.G., Z.B.Y., H.B.Z. and Y.K. collected the adiabatic temperature data from the direct measurement. Y.H.G. and S.K.Y collected and analyzed the infrared data. D.H.W. and Y.H.G. did the molecular dynamics simulation with help of Y.L.L., J.W. and Y.Liu. Y.H.G. analyzed the simulation results. Y.H.G. and F.X.H. wrote the paper with input from all coauthors. Y.Lin, Y.Z.C., J.R.S. and T.Y.Z. contributed to discussing and revising the paper.

## Competing interests

The authors declare no competing interests.
