## [Peer Review File · Nature Communications]

REVIEWER COMMENTS

Reviewer #1 (Remarks to the Author):

The authors report a colossal barocaloric effect in a plastic crystal, i.e., $\text{CH}_3(\text{CH}_2)_9\text{NH}_2\text{Cl}$, characteristic of a chained structure. The material exhibits remarkable barocaloric performances, including low driving pressure, small thermal hysteresis, large entropy change, and adiabatic temperature change. This is a start-of-the-art advance since the original discovery of colossal barocaloric effects in plastic crystals consisting of small molecules. In addition, the authors also explored the underlying physical mechanism using molecular dynamics simulations. Overall, this is an interesting work and represents the emerging direction of the barocaloric materials research community. I strongly recommend it be published in the journal, however, a few issues have to be resolved beforehand, as below.

- 1) The authors highlighted “two dimensions” and “vdw” in the title of the manuscript. However, these terms haven’t been well established and are not relevant to the physical importance. The layered structure seems confined in the bc plane. In the bc plane, the chains are connected via the ionic bonds as well as hydrogen bonds. Along the a direction, there seems no strong coupling (Fig. 1c). In addition, vdw interaction is quite common in organic solids so I didn’t see the uniqueness here.
- 2) The measurements of the adiabatic temperature change must be well-calibrated. I would suggest the authors try common chemicals like NaCl and Al_2O_3 at first, which would provide a benchmark. Looking at Fig. 4, Fig. S8 and Fig. S9, the pressure dependence of the adiabatic temperature change is weird. Why 6.1 K at 0.3 Gpa, but 6.8 K at 0.2 Gpa? At 0.1 Gpa, the adiabatic temperature change is 6 K, not 11 K as mentioned in the abstract. In addition, the adiabatic temperature change estimated using the entropy change data is quite different from the measured one. If the authors couldn’t explain such a difference, I would suggest removing the estimated one.
- 3) The expansion along the c axis at the phase transition seems disconnected from the molecular motions. The curved molecules are rigid in the low-temperature phase and become foldable in the high-temperature phase. To me, the effective length of the molecules should become shorter instead of longer. In this case, I would suggest the authors calculate the projected length of the molecules along the c axis. With an order parameter, the authors should be able to reproduce the phase transition in their simulations.
- 4) As one of the appealing merits, the pressure sensitivity was not well explained.
- 5) The full information of the single crystal structural refinement has to be given. The lattice constant a and b must be plotted in Fig. 5C.
- 6) Gpa, bar and kbar all appeared. Please keep consistent.
- 7) The thermal hysteresis was claimed to be 5.9 K, but much larger in the phase diagram.
- 8) The information listed in Table 1 is a little bit arbitrarily selected as barocaloric materials with entropy change higher than $100 \text{ J kg}^{-1} \text{ K}^{-1}$ are quite many at present.

Reviewer #2 (Remarks to the Author):

The authors investigate the barocaloric response of $(\text{CH}_3-(\text{CH}_2)_9)_2\text{NH}_2\text{Cl}$, which is a new compound of the so-called two-dimensional van der Waals type. It consists of organic chains linked by inorganic ions, where organic chains facing each other are stabilized by van der Waals interactions and show strong disorder (pseudomelting) consisting of emergence and mobility of different conformations along the chain. Such dynamics (and the structure of the organic chain arrangement) is identical to that exhibited by 2-dimensional hybrid pseudoperovskites, but in this case the perovskite structure is lacking. The novelty requirement for publication in Nature Commun. is therefore met for the novelty of the compound. Beyond that, the work is of merit because the barocaloric response, despite not showing an impressive jump compared to previous reported materials, under moderate pressure changes is excellent. Moreover, the set of reported experiments is very complete, including a wide range of experimental techniques such as calorimetry, single crystal and powder X-ray diffraction, FTIR and direct measurements of (nearly)adiabatic temperature changes, along with molecular dynamics simulations.

- I understand that the compound has not been synthesized before, but the literature on this material family is quite extensive and the text is somewhat ambiguous in this sense. I strongly suggest the authors to make it clearer by including some sentence like “unprecedented”, “for the first time”, etc.

- I do not understand why the authors call the material a “plastic crystal”, as they do not have the dynamical properties exhibited by plastic crystals: basically, molecular orientational disorder but also soft mechanical behavior associated with some diffusion. Maybe I am missing something but please explain better or remove this statement.

- Lines 71-73 read: “Meanwhile, large geometry incompatibility of two phases during phase transition related to the drastic symmetry breaking/restoration of lattice tends to cause large hysteresis³¹.” The cited reference 31 precisely gives a counter example, please give other references.

- In several places, entropy S is referred to entropy changes associated to certain processes and not to the absolute entropy. Therefore, the notation should be changed (in the y-label itself) to avoid mistaken quantities or ambiguities. For instance: Fig. 1e refers to transition entropy change.

- Fig. 1a: Mention that X-rays are at atmospheric pressure.

- DSC measurement shown in Fig. 1d is conducted at 0.1 K/min. This rate is unusually low for this technique. Despite it may be meaningful to establish the endothermic transition temperature, it may be very biased when trying to determine the hysteresis because the latter can be dramatically affected by the scanning rate. Indeed, DSC measurements performed at faster rates (at atmospheric pressure or high pressure) display a transition with a larger hysteresis (1 K/min renders a hysteresis of 1 K). I suspect that the hysteresis can be even larger when going to higher rates, which then could entail a decrease in the reversible performance. Please revise and mention such dependence.

- Lines 249-251 read “The superior ΔT_r at low pressure can be revealed by comparing the dC10Cl with all the reported materials measured by direct measurement method (Supplementary Fig. S10)”. I find this comparison unfair. The comparison should include ΔT obtained by other methods, too, otherwise it is somewhat unfair and biased. Of course the particular methodology in each case must be mentioned.

- Fig. 4: the image strongly seems to show a powder sample, and in the schematic of the setup it is also displayed in powder form. So saying here single crystal is confusing. Please revise.
- Powder x-ray diffraction is abbreviated as Pw-XRD. Please use PXRD or XRPD instead, as this is the most used, standard notation.
- In Fig. 5c the authors show thermal expansion. Why not to include it in their calculations of the barocaloric effects? At first sight it can be seen that this would entail a further enhancement of the isothermal entropy changes.
- Table 1: Why 1-Cl-ada and 1-Br-ada are not included in the cyan background, i.e. as plastic crystals? On the other hand, many compounds are missing, please revise and extend.
- Lines 479-480 read “[...] the 3D plastic crystals exhibit limited volume change across the phase transition [...]” but volume changes in these materials can reach up to 10%, which is also huge and comparable to the current compound. I suggest to soften the mentioned sentence.
- In lines 320-321 the authors mention the Clausius-Clapeyron but I could not find any calculation to check the agreement of this equation with their own experimental data. This might be particularly relevant in this case because their high-pressure calorimeter uses gas as a pressure-transmitting fluid. In some compounds it might happen that some of the gas molecules penetrate inside the compound, yielding to a different system showing a dT/dp which does not correspond to the pure compound. Using the Clausius-Clapeyron equation would help to give reliability on this feature. Please include.

Following are the point-by-point responses to the reviewers:

Reviewers' comments (Blue) and Responding to Reviewers' Comments (Black):

For Reviewer #1

The authors report a colossal barocaloric effect in a plastic crystal, i.e., $\text{CH}_3(\text{CH}_2)_9)_2\text{NH}_2\text{Cl}$, characteristic of a chained structure. The material exhibits remarkable barocaloric performances, including low driving pressure, small thermal hysteresis, large entropy change, and adiabatic temperature change. This is a start-of-the-art advance since the original discovery of colossal barocaloric effects in plastic crystals consisting of small molecules. In addition, the authors also explored the underlying physical mechanism using molecular dynamics simulations. Overall, this is an interesting work and represents the emerging direction of the barocaloric materials research community. I strongly recommend it be published in the journal, however, a few issues have to be resolved beforehand, as below.

Thanks so much for the valuable comments from Reviewer #1.

1. The authors highlighted “two dimensions” and “vdw” in the title of the manuscript. However, these terms haven’t been well established and are not relevant to the physical importance. The layered structure seems confined in the bc plane. In the bc plane, the chains are connected via the ionic bonds as well as hydrogen bonds. Along the a direction, there seems no strong coupling (Fig. 1c). In addition, vdw interaction is quite common in organic solids so I didn’t see the uniqueness here.

We appreciate the suggestion of the reviewer, which conduces us to further clarifying the structure characteristics of this material system.

As described in the main text, for the system of $(\text{CH}_3-(\text{CH}_2)_9)_2\text{NH}_2\text{Cl}$ plastic crystal, there exists anisotropic interaction containing the weak interlayer and strong intralayer interaction. Concretely, as shown in Fig. 1 in manuscript (shown below here), the interlayer interaction along the *c* axis arises from the vdW force between the CH_3 groups at the end of organic chains in adjacent molecular layers, while the intralayer interaction is dominated by the anion-cation interaction, between the N^+ and Cl^- , and the $\text{N}-\text{H}\dots\text{Cl}$ hydrogen bond interaction in the *ab* plane. Within the intralayer, it can be considered that the anion-cation interaction maintains the compact arrangement of molecules in the *ab* plane, and the hydrogen bond interaction further determines the orientation of organic chains. Therefore, based on the vdW interaction along the *c* axis, and the ionic interaction and hydrogen bond force in the *ab* plane, the two-dimensional vdW structure then has been established in system of plastic crystal $(\text{CH}_3-(\text{CH}_2)_9)_2\text{NH}_2\text{Cl}$ (dC_{10}Cl for short).

To demonstrate that more strongly, we calculate the reduced density gradient (RDG) related to the

electron density, which can reveal the intensity of molecular interaction. The calculation results based on the density functional theory (DFT) and the Multiwfn program^{R1} are shown in Fig. R1-R5.

[Fig. 1 For $((\text{CH}_3-(\text{CH}_2)_9)_2\text{NH}_2)\text{Cl}$. **a** Crystalline structure of $((\text{CH}_3-(\text{CH}_2)_9)_2\text{NH}_2)\text{Cl}$ at 300 K, where the unit cell is marked by black frame. **b** Molecular structure of $((\text{CH}_3-(\text{CH}_2)_9)_2\text{NH}_2)\text{Cl}$. **c** N-H...Cl hydrogen bond interaction in $((\text{CH}_3-(\text{CH}_2)_9)_2\text{NH}_2)\text{Cl}$, where partial carbon and hydrogen atoms are omitted.]

Considering the complexity arising from the multiple-atom-system, the interlayer and intralayer interaction were researched based on $1.5 \times 1.5 \times 1$ double-layer and $2 \times 2 \times 0.5$ single-layer molecular models, respectively. For two-layer model, RDG of $d\text{C}_{10}\text{Cl}$ is shown in Fig. R1. The downward spike around $\text{sign}(\lambda_2)\rho \sim -0.005$ indicates the weak vdW attractive force, which can be further depicted in the real space with the specific coloring rule in Fig. R2. As shown in Fig. R2, the vdW interaction of $\rho \sim 0$ and $\lambda_2 \sim 0$ would be depicted in green, and the stronger attractive force can be depicted in color closer to blue. Thus, the vdW interaction can be depicted in Fig. R3, and the framed part exhibits the soft interlayer vdW force, which is weaker than the vdW interaction of interchains of each layer.

Fig. R1 RDG of dC₁₀Cl based on the two-layer molecular model.

Fig. R2 Corresponding colors of the interaction species.

Fig. R3 The vdW force in two-layer molecular model.

Moreover, based on the single-layer molecular model, the intralayer interaction can be well illustrated. As shown in Fig. R4, the downward spike around $\text{sign}(\lambda_2)\rho \sim -0.005$ still exists. It refers to the intralayer interchain vdW force, which is also able to be seen in Fig. R3. Located closely left of the vdW spike, the downward spike at $\text{sign}(\lambda_2)\rho \sim -0.02$ indicates the relatively strong hydrogen bond interaction in dC₁₀Cl system, which is further depicted with cyan electron cloud between N atoms and Cl atoms in Fig. R5. Moreover, the downward spike in the range of $\text{sign}(\lambda_2)\rho \sim -0.24$ can indicate the ionic attractive force, as shown in Fig. R6 and Fig. R7, where the blue electron cloud around the Cl atoms reveals the strong intralayer ionic interaction. It should be noted that the ionic interaction is present along both the *a* and *b* axis in *ab* plane, and ensures the close arrangement of ions in intralayer *ab* plane.

Therefore, combining the RDG results of two-layer and single-layer model, the anisotropic interaction in dC₁₀Cl can be well established, where interlayer weak vdW force, and intralayer strong ionic and hydrogen bond interaction contribute to the 2D layered structure in dC₁₀Cl.

The above RDG analysis for clarifying the interaction in (C₁₀H₂₁)₂NH₂Cl (dC₁₀Cl) has been provided in Supplementary Note 1.

Fig. R4 RDG of dC₁₀Cl based on the single-layer molecular model.

Fig. R5 The hydrogen bond interaction in single-layer molecular model.

Fig. R6 The ionic interaction perpendicular to the *ac* plane in single-layer molecular model.

Fig. R7 The ionic interaction perpendicular to the *bc* plane in single-layer molecular model.

2. The measurements of the adiabatic temperature change must be well-calibrated. I would suggest the authors try common chemicals like NaCl and Al₂O₃ at first, which would provide a benchmark. Looking at Fig. 4, Fig. S8 and Fig. S9, the pressure dependence of the adiabatic temperature change is weird. Why 6.1 K at 0.3 Gpa, but 6.8 K at 0.2 Gpa? At 0.1 Gpa, the adiabatic temperature change is 6 K, not 11 K as mentioned in the abstract. In addition, the adiabatic temperature change estimated using the entropy change data is quite different from the measured one. If the authors couldn't explain such a difference, I would suggest removing the estimated one.

Thanks so much for the advice.

To make the measured adiabatic temperature change (ΔT_{ad}) reliable, we indeed chose NaCl as a benchmark and performed the direct measurement of ΔT_{ad} before each measurement for present dC₁₀Cl. A typical result is given in Fig. R8. At ~296 K, a pressure of 0.1 GPa was applied on NaCl and then released. The temperature error is within 0.1 K. One can see there are no temperature change detected during either pressurization or depressurization process, which illustrated the well-calibrated ΔT_{ad} for the present dC₁₀Cl.

With regard to the pressure dependence of the measured ΔT_{ad} during depressurization, as the reviewer mentioned, the non-linear relation and irregularity therein completely derive from the defect of our device. As shown in Fig.4a and contrastive illustration in Fig.4b, the sharp increase of temperature in the first process (I) indicates the thermal effect produced by approximately adiabatic pressurization to 0.1 GPa. However, the applied pressure cannot be fully maintained but spontaneously release some due to our device, which corresponds to the second process (II), where the heat exchange between the heated sample and environment leads to the cooling trend. Then, in the third process (III) of depressurization, the actual starting point of pressure is lower than the initial set point and possesses random character due to unavoidable spontaneous release in the II process, which thus results in the irregular pressure dependence of the detected ΔT_{ad} during depressurization.

However, the temperature increase during pressurization, schematically illustrated as the process I in Fig.4a, guarantees the accuracy of applied pressure value, and hence the pressure dependence of the ΔT_{ad} is credible, noting the repeatable ΔT_{ad} value ($\Delta T_{ad} \sim 11\text{K}$ under 0.1GPa) on the upward branch during cyclic measurements (Fig.4c). Therefore, the reliability of reversible ΔT_{ad} can be well determined.

In addition, the magnitude of reversible ΔT_{ad} (ΔT_r) estimated using the entropy change data by quasi-direct method ($\Delta T_{ad} \sim 9.0\text{K}$ under 0.1GPa, Fig.2f of main text) is actually similar to the one from the direct measurement (Fig.4c, $\Delta T_{ad} \sim 11\text{K}$ under 0.1GPa) taking into account possible errors caused by different methods. All these illustrate the giant BCE achieved in the present dC₁₀Cl of 2D plastic crystals.

Fig. R8 Direct measurement of adiabatic temperature change on NaCl upon pressurization and depressurization, where the pressure ranges from 0 to 0.1GPa.

[Fig. 4 The direct measurement of adiabatic temperature change ΔT_{ad} driven by pressure in $dC_{10}Cl$. **a** ΔT_{ad} induced by pressure of 0.1 GPa. **b** Schematic illustration of temperature change and pressure evolution during the adiabatic test. **c** Cycle measurements of ΔT_{ad} induced by pressure of 0.1 GPa. **d** Schematic diagram of direct measurement of ΔT_{ad} , where the sample's image in single crystal is shown at top right.]

Fig. R8 has been provided in Supplementary materials, Fig. S8.

3. The expansion along the c axis at the phase transition seems disconnected from the molecular motions. The curved molecules are rigid in the low-temperature phase and become foldable in the high-temperature phase. To me, the effective length of the molecules should become shorter instead of longer. In this case, I would suggest the authors calculate the projected length of the molecules along the c axis. With an order parameter, the authors should be able to reproduce the phase transition in their simulations.

As the reviewer mentioned here, it is intriguing and significant to clarify the interrelation between the expansion of lattice along the c axis and the dynamic motions of organic chains during the phase transition. As shown in Fig. 6 of the manuscript, at the ordered state, the organic chains arrange regularly and the extension direction of organic chains forms an angle of 45° to the c axis. At the disordered state, the organic chains should get twisted and shorter intuitively, however, the orientations of organic chains are disordered and exhibit the trend of smaller angle to the c axis. Therefore, the expansion along the interlayer direction of c axis appears to be large.

Furthermore, the weaker vdW force between chains along the c axis, which can be proved with CH_3 stretching vibration blueshift in IR spectra (Fig. 8e in the manuscript), also results in enlarged distance along the c axis between organic chains. Ultimately, the expansion along c can be realized.

[Fig. 6 MD simulation on the $(CH_3-(CH_2)_9)_2NH_2Cl$.]

[Fig. 8 Temperature-variable infrared spectra of $(\text{CH}_3-(\text{CH}_2)_9)_2\text{NH}_2\text{Cl}$.]

4. As one of the appealing merits, the pressure sensitivity was not well explained.

According to the suggestion of the reviewer, we have added the concrete explanation of pressure sensitivity into the manuscript. See Page 15, line 323.

Based on the Clausius-Clapeyron relation ($dT_s/dP = \Delta V/\Delta S$), the large entropy change across the phase transition tends to induce the low pressure-sensitivity of phase transition. Significantly, in the two-dimensional van-der-Waals plastic crystal dC_{10}Cl , the volume change of phase transition appears to be colossal, due to the weak interlayer vdW interaction. In this case, the large pressure sensitivity of phase transition with colossal entropy change can be accomplished with the compensation of the large volume expansion. Therefore, the coexistence of colossal entropy change and magnificent volume change during the phase transition contributes to the superior barocaloric effect reversibly driven by low pressure.

5. The full information of the single crystal structural refinement has to be given. The lattice constant a and b must be plotted in Fig. 5C.

According to the suggestion of the reviewer, we have supplemented the information of the single crystal structural refinement in the Supplementary Information (See Page 17). And we have completed the temperature dependence of lattice constant a and b in manuscript, as shown in Fig.

R9 here.

Fig. R9 (Fig. 5c) The temperature dependence of cell volume and lattice parameters in $dC_{10}Cl$.

6. Gpa, bar and kbar all appeared. Please keep consistent.

We appreciate the kind reminder of the reviewer. We have revised the unit of pressure appeared in the manuscript, and all were unified as GPa and MPa.

7. The thermal hysteresis was claimed to be 5.9 K, but much larger in the phase diagram.

The thermal hysteresis of 5.9 K was measured with temperature rate of 0.1 K/min (Fig. 1d), while the counterpart in phase diagram was acquired with 1 K/min (Fig. 2b), which gives rise to the difference. For clarity, the temperature ramping rate has been clearly written in the corresponding figure captions.

8. The information listed in Table 1 is a little bit arbitrarily selected as barocaloric materials with entropy change higher than 100 J kg⁻¹ K⁻¹ are quite many at present.

According to the suggestion of the reviewer, we have added the omissive colossal barocaloric materials with solid-solid phase transition (entropy change over 100 J kg⁻¹ K⁻¹) into the Table 1 of manuscript, including copolymer PEG10000/PET15000^{R2}, 1-adamantanol and 2-methyl-2-adamantanol^{R3}, and NH₄SCN^{R4}. And now we believe that the Table 1 for comparison of colossal barocaloric materials with solid-solid phase transition exhibiting entropy change larger than 100 J kg⁻¹ K⁻¹ should be complete. See the revised Table 1 in the main text.

References:

- R1. Lu, T. & Chen, F. Multiwfn: A multifunctional wavefunction analyzer. *J. Comput. Chem.* **33**, 580–592 (2012).
- R2. Yu, Z. *et al.* Colossal barocaloric effect achieved by exploiting the amorphous high entropy of solidified polyethylene glycol. *NPG Asia Mater.* **14**, 96 (2022).
- R3. Salvatori, A. *et al.* Colossal barocaloric effects in adamantane derivatives for thermal management. *APL Mater.* **10**, 111117 (2022).
- R4. Zhang, Z. *et al.* Thermal batteries based on inverse barocaloric effects. *Sci. Adv.* **9**, eadd0374 (2023).

For Reviewer #2

The authors investigate the barocaloric response of $(\text{CH}_3 - (\text{CH}_2)_9)_2\text{NH}_2\text{Cl}$, which is a new compound of the so-called two-dimensional van der Waals type. It consists of organic chains linked by inorganic ions, where organic chains facing each other are stabilized by van der Waals interactions and show strong disorder (pseudomelting) consisting of emergence and mobility of different conformations along the chain. Such dynamics (and the structure of the organic chain arrangement) is identical to that exhibited by 2-dimensional hybrid pseudoperovskites, but in this case the perovskite structure is lacking. The novelty requirement for publication in Nature Commun. is therefore met for the novelty of the compound. Beyond that, the work is of merit because the barocaloric response, despite not showing an impressive jump compared to previous reported materials, under moderate pressure changes is excellent. Moreover, the set of reported experiments is very complete, including a wide range of experimental techniques such as calorimetry, single crystal and powder X-ray diffraction, FTIR and direct measurements of (nearly)adiabatic temperature changes, along with molecular dynamics simulations.

Thanks so much for the valuable comments from Reviewer #2.

1. I understand that the compound has not been synthesized before, but the literature on this material family is quite extensive and the text is somewhat ambiguous in this sense. I strongly suggest the authors to make it clearer by including some sentence like “unprecedented”, “for the first time”, etc.

Thanks so much for the kind advice.

The $(\text{CH}_3 - (\text{CH}_2)_9)_2\text{NH}_2\text{Cl}$ materials and single crystals were synthesized here for the first time, then the refined structure was investigated and the two-dimensional van-der-Waals structural character can be revealed. As suggested by the reviewer, we have included the sentence like “for the first time” to make it clear. See Page 6, line 126.

2. I do not understand why the authors call the material a “plastic crystal”, as they do not have the dynamical properties exhibited by plastic crystals: basically, molecular orientational disorder but also soft mechanical behavior associated with some diffusion. Maybe I am missing something but please explain better or remove this statement.

As for the question of the reviewer, we now explain the term of “plastic crystal” more carefully. In 1961, Timmermans firstly propose the term of “plastic crystals” to describe the specific materials, where molecular systems are formed with nearly globular molecules that enable orientational degrees of freedom and consequent plasticity^{R5}. Subsequently, ionic plastic crystals were also explored, consisted of anionic and cationic species^{R6,R7}. Moreover, large molecules with disk-like and rod-like (long alkyl chains for instance) species, no longer confined to the globular shape, were also discovered to form plastic crystal phases^{R8,R9}. In the context of diverse plastic crystal systems, we discovered the materials in this article as ionic plastic crystal with rod-like cationic species. And the molecular dynamic character and the self-diffusion plastic mechanic behavior in this system will be elucidated below.

Firstly, we have demonstrated that in the plastic crystal phase of compound $(\text{CH}_3 - (\text{CH}_2)_9)_2\text{NH}_2\text{Cl}$,

there exists the conformation disorder of the organic chains, which refers to the disorderly twisting of organic chains and essentially the reorientation of partial groups in organic chains. Secondly, we discovered that under a pressure as small as 100 MPa, the $(\text{CH}_3-(\text{CH}_2)_9)_2\text{NH}_2\text{Cl}$ samples at plastic crystal phase above transition temperature (T_s) can stick together (right one of Fig.R10) but cannot at room temperature below the T_s (left one of Fig.R10), which substantiates the severe plastic flow and self-diffusion^{R7,R9,R10}.

Therefore, with the characteristics of plastic crystal as illustrated above and the two-dimensional structure proved by SC-XRD, it is reasonable to call the material two-dimensional vdW plastic crystals.

The corresponding explanation has been added into the Supplementary Note 2 of Supplementary information.

Fig. R10 The image of $(\text{CH}_3-(\text{CH}_2)_9)_2\text{NH}_2\text{Cl}$ samples after the pressurization of 100 MPa at room temperature (left) and 330 K (right).

3. Lines 71-73 read: “Meanwhile, large geometry incompatibility of two phases during phase transition related to the drastic symmetry breaking/restoration of lattice tends to cause large hysteresis³¹.” The cited reference 31 precisely gives a counter example, please give other references.

Thanks so much for the kind reminder of the reviewer. We have replaced the reference 31 here with other references (see Ref.[31,32,33] in the revised manuscript). In these references, specifically, Cui et al. performed the high-throughput approach and verified the clear relationship between the hysteresis and the crystal symmetry and geometric compatibilities of two phases in Ni-Ti-Cu alloys^{R11}. And Zarnetta et al. further developed the near-zero thermal hysteresis alloy $\text{Ti}_{50.2}\text{Ni}_{34.4}\text{Cu}_{12.3}\text{Pd}_{3.1}$ based on above relationship^{R12}. Moreover, Liang et al. discovered that the hysteresis of a correlated oxide system $\text{V}_{1-x}\text{W}_x\text{O}_2$ can be directly tuned by adjusting the lattice compatibility between transformed and parent phases^{R13}.

4. In several places, entropy S is referred to entropy changes associated to certain processes and not to the absolute entropy. Therefore, the notation should be changed (in the label itself) to avoid mistaken quantities or ambiguities. For instance: Fig. 1e refers to transition entropy change.

According to the suggestion of the reviewer, we have changed the notations to entropy change ΔS in Fig. 1e of the manuscript and also Supplementary Fig. S5 of the supplementary information.

5. Fig. 1a: Mention that X-rays are at atmospheric pressure.

According to the reminder of the reviewer, we have added the test condition of atmospheric pressure into the description of single crystal and powder X-ray diffraction measurements. See Page 6 line 131 and page 12 line 270.

6. DSC measurement shown in Fig. 1d is conducted at 0.1 K/min. This rate is unusually low for this technique. Despite it may be meaningful to establish the endothermic transition temperature, it may be very biased when trying to determine the hysteresis because the latter can be dramatically affected by the scanning rate. Indeed, DSC measurements performed at faster rates (at atmospheric pressure or high pressure) display a transition with a larger hysteresis (1 K/min renders a hysteresis of 1 K). I suspect that the hysteresis can be even larger when going to higher rates, which then could entail a decrease in the reversible performance. Please revise and mention such dependence.

We agree with the reviewer that the hysteresis can be larger when adopting higher temperature rates in testing. In this article, we initially gave the thermal hysteresis (5.9 K) of phase transition with low temperature rate of 0.1 K/min in Fig. 1, which excludes the extrinsic factors as much as possible and acquires the intrinsic hysteresis. Then the pressure-variable DSC measurements were performed at temperature rate of 1 K/min, revealing a somewhat larger hysteresis (8 K) independent of pressure (see Fig. 2b).

According to the suggestion of the reviewer, we performed the DSC measurements at a higher temperature rate of 2 K/min, which is the upper limit our instrument can reach (SETARAM μ DSC7 evo microcalorimeter). Then a wider thermal hysteresis ~ 10 K was measured. As a result, the pressure driving the maximum reversible entropy change ($400 \text{ J kg}^{-1} \text{ K}^{-1}$) slightly enhanced to 0.1 GPa, noting this value is 0.08 GPa in the case of 1 K/min.

As for the intensified hysteresis at higher temperature rate, it can be reasonably explained that the heat exchange between the samples and detector can be more insufficient at higher temperature rate, which induces an artificial hysteresis effect and consequently extrinsic irreversible barocaloric effect. In this work of exploring the novel barocaloric materials, we actually concentrate on the intrinsic hallmark of phase transition and barocaloric effect. Therefore, the thermal measurements were performed with 0.1 K/min and 1 K/min as mentioned above.

Moreover, at the reminder of the reviewer, we added the corresponding statement into the manuscript, and mentioned the possible extrinsic effect on barocaloric effect if a high temperature rate was adopted. See page 9 line 186.

7. Lines 249-251 read “The superior ΔT_r at low pressure can be revealed by comparing the dC10Cl with all the reported materials measured by direct measurement method (Supplementary Fig. S10)”. **I find this comparison unfair.** The comparison should include ΔT_r obtained by other methods, too, otherwise it is somewhat unfair and biased. Of course the particular methodology in each case must be mentioned.

We appreciate the suggestion of the considerate reviewer. According to the suggestion, we added the reversible adiabatic temperature change ΔT_r by non-direct methods for some representative colossal barocaloric materials with phase transition entropy change near or above $100 \text{ J kg}^{-1} \text{ K}^{-1}$.

The comparison figure is shown in Fig. R11 here (Fig. S11 in the supplementary information). And corresponding discussion has been revised in the main text, see page 11 line 253.

Fig. R11 (a) The directly measured reversible adiabatic temperature change ΔT of our $dC_{10}Cl$ compared to other barocaloric materials we can find in the literatures^{R14-R22}. (b) The reversible ΔT of some representative colossal barocaloric materials with phase transition entropy change near or above $100 \text{ J kg}^{-1} \text{ K}^{-1}$ by quasi-direct method^{R23-R29}.

8. Fig. 4: the image strongly seems to show a powder sample, and in the schematic of the setup it is also displayed in powder form. So saying here single crystal is confusing. Please revise.

The sample image in Fig. 4d shows the transparent thin sheet form of the samples, while the image of the setup shows the samples in powder form just for illustrative purpose. To be clear, we have further explained that in the caption (Fig.4d. Schematic diagram of direct measurement of ΔT_{ad} , where the sample's image is for indication only, and the real image of single crystal sample is shown at top right). See page 12 line 262. Moreover, to be clear, we show the optical microscopic image of the samples as below, see Fig. R12.

Fig. R12 Optical microscopic image of the $dC_{10}Cl$ single crystal samples.

9. Powder x-ray diffraction is abbreviated as Pw-XRD. Please use PXRD or XRPD instead, as this is the most used, standard notation.

According to the suggestion of the reviewer, we have modified “Pw-XRD” to “PXRD” in the manuscript. See the highlighted revisions by yellow.

10. In Fig. 5c the authors show thermal expansion. Why not to include it in their calculations of the barocaloric effects? At first sight it can be seen that this would entail a further enhancement of the isothermal entropy changes.

As the reviewer mentioned, the relationship between the volume change and entropy change should be explained. As for the barocaloric materials, the volume change arises from the interplay between the structure and additional degrees of freedom, such as conformational ordering in our work^{R30}. Likewise, the entropy change originates from the structure change and the order-disorder transition for the degrees of freedom. Therefore, the relationship between the volume change and entropy change can be established utilizing the underlying phase transition mechanism involving the structure evolution and degrees of freedom.

As for the system in our work, the colossal volume expansion actually dictates the substantial free space of organic chains, which enables the severe conformational disorder and hence induces the colossal entropy change during the phase transition.

The idea of above description has been added to the main text, see Page 15 line 317.

11. Table 1: Why 1-Cl-ada and 1-Br-ada are not included in the cyan background, i.e. as plastic crystals? On the other hand, many compounds are missing, please revise and extend.

We appreciate the reminder of the reviewer. According to the suggestion of the reviewer, we have included 1-Cl-ada and 1-Br-ada in the cyan background. Moreover, we have added other colossal barocaloric compounds with solid-solid phase transition (having entropy change larger than 100 J kg⁻¹K⁻¹) into Table 1 for completeness, which are copolymer PEG10000/PET15000^{R2}, 1-adamantanol and 2-methyl-2-adamantanol^{R3}, and NH₄SCN^{R4}. See the revised Table 1 in the main text.

12. Lines 479-480 read “[...] the 3D plastic crystals exhibit limited volume change across the phase transition [...]” but volume changes in these materials can reach up to 10%, which is also huge and comparable to the current compound. I suggest to soften the mentioned sentence.

We appreciate the reminder of the reviewer. According to the suggestion, we have revised the corresponding words and softened the sentence. See “the 3D plastic crystals exhibit relatively limited volume change across the phase transition” on page 25 line 489.

13. In lines 320-321 the authors mention the Clausius-Clapeyron but I could not find any calculation to check the agreement of this equation with their own experimental data. This might be particularly relevant in this case because their high-pressure calorimeter uses gas as a pressure-transmitting fluid. In some compounds it might happen that some of the gas molecules penetrate inside the compound, yielding to a different system showing a dT/dp which does not correspond to the pure compound. Using the Clausius-Clapeyron equation would help to give reliability on this feature. Please include.

We appreciate the suggestion of the considerate reviewer. We now consider the calculation based on the Clausius-Clapeyron relation ($dT_s/dP = \Delta V/\Delta S$) and conduct the comparison of

experimental and calculated dT_s/dP . Table R1 summarizes the dT_s/dP (pressure sensitivity of phase transition temperature) experimentally measured by P-DSC and the one calculated from Clausius-Clapeyron equation for present $dC_{10}Cl$ and many other materials. One can note the non-negligible deviations between the experimental and calculated dT_s/dP for most materials. The difference ratio can be over 30% for the reported TRIS and our $dC_{10}Cl$. Then there seems to be a general rule that the measured dT_s/dP is smaller than the one from C-C equation. Just like the mentioned phenomenon by reviewer, the key reason might be the penetration of pressure-transmitting gas molecules into the compound, which makes the real pressure exerting on samples lower than the setting value. Hence, a lower dT_s/dP was experimentally detected.

Moreover, most materials listed in Table R1 are newly discovered and synthesized, their full physical properties are not yet known, and the ability penetrating by gas molecules is not the same from one material to another, and may vary considerably, so behaving different deviation of dT_s/dP compared to the one from its C-C equation for different materials.

The present 2D plastic crystals of $dC_{10}Cl$ are also newly discovered and synthesized materials. We found that the $dC_{10}Cl$ exhibits high diffusivity in the plastic crystal state above T_s , and can easily stick under a small pressure, but its gas-penetrating ability is not clear to us, and the related properties are yet to be studied in depth.

The corresponding explanation has been included in Supplementary Note 3 of Supplementary information.

Table R1. The comparison of dT_s/dP experimentally measured by P-DSC and the one from Clausius-Clapeyron equation. ΔS and ΔV denote the entropy change and volume change, respectively, across phase transition. dT_s/dP (experimentally) is the experimentally measured by P-DSC. dT_s/dP (C-C equation $\Delta V/\Delta S$) is the one calculated by Clausius-Clapeyron relation.

Compounds	$ \Delta S $ (J kg ⁻¹ K ⁻¹)	dT_s/dP (K GPa ⁻¹) (experimentally)	ΔV (E-5 m ³ kg ⁻¹)	dT_s/dP (K GPa ⁻¹) (C-C equation $\Delta V/\Delta S$)	Error(+-%)	Ref.
$dC_{10}Cl$	400	190	11.9	298	36.2	--
NPG	384	133	4.6	120	10.8	[R31, R32]
PG	485	79	5.1	105	24.8	[R27, R33]
TRIS	682	37	3.7	54	31.5	[R27, R34]
AMP	632	64	4.6	73	12.3	[R27, R33]
1-Cl-ada	132	270	4.7	356	24.2	[R24]
1-Br-ada	102	333	4	392	15.1	[R24]
$Fe_3(bntrz)_6$ (tcnset) ₆	80	250	2.1	263	4.9	[R23]

References:

- R5. Timmermans, P. J. Plastic crystals: a historical review. *J. Phys. Chem. Solids* **18**, 1–8 (1961).
R6. Zhu, H., MacFarlane, D. R., Pringle, J. M. & Forsyth, M. Organic Ionic Plastic Crystals as Solid-State Electrolytes. *Trends in Chemistry* **1**, 126–140 (2019).
R7. Harada, J. *et al.* Directionally tunable and mechanically deformable ferroelectric crystals from rotating polar globular ionic molecules. *Nat. Chem.* **8**, 946–952 (2016).

- R8. Laschat, S. *et al.* Discotic Liquid Crystals: From Tailor-Made Synthesis to Plastic Electronics. *Angew. Chem., Int. Ed.* **46**, 4832–4887 (2007).
- R9. Das, S., Mondal, A. & Reddy, C. M. Harnessing molecular rotations in plastic crystals: a holistic view for crystal engineering of adaptive soft materials. *Chem. Soc. Rev.* **49**, 8878–8896 (2020).
- R10. Mondal, A. *et al.* Metal-like Ductility in Organic Plastic Crystals: Role of Molecular Shape and Dihydrogen Bonding Interactions in Aminoboranes. *Angew. Chem., Int. Ed.* **59**, 10971–10980 (2020).
- R11. Cui, J. *et al.* Combinatorial search of thermoelastic shape-memory alloys with extremely small hysteresis width. *Nat. Mater.* **5**, 286–290 (2006).
- R12. Zarnetta, R. *et al.* Identification of Quaternary Shape Memory Alloys with Near-Zero Thermal Hysteresis and Unprecedented Functional Stability. *Adv. Funct. Mater.* **20**, 1917–1923 (2010).
- R13. Liang, Y. G. *et al.* Tuning the hysteresis of a metal-insulator transition via lattice compatibility. *Nat. Commun.* **11**, 3539 (2020).
- R14. Yuce, S. *et al.* Barocaloric effect in the magnetocaloric prototype Gd₅Si₂Ge₂. *Appl. Phys. Lett.* **101**, 071906 (2012).
- R15. Bocca, J. R. *et al.* Giant barocaloric effect in commercial polyurethane. *Polymer Testing* **100**, 107251 (2021).
- R16. Carvalho, A. M. G., Imamura, W., Usuda, E. O. & Bom, N. M. Giant room-temperature barocaloric effects in PDMS rubber at low pressures. *Eur. Polym. J.* **99**, 212–221 (2018).
- R17. Usuda, E. O., Bom, N. M. & Carvalho, A. M. G. Large barocaloric effects at low pressures in natural rubber. *Eur. Polym. J.* **92**, 287–293 (2017).
- R18. Matsunami, D., Fujita, A., Takenaka, K. & Kano, M. Giant barocaloric effect enhanced by the frustration of the antiferromagnetic phase in Mn₃GaN. *Nat. Mater.* **14**, 73–78 (2015).
- R19. Liu, Y. *et al.* Large barocaloric effect in intermetallic La_{1.2}Ce_{0.8}Fe₁₁Si₂H_{1.86} materials driven by low pressure. *NPG Asia Mater.* **14**, 30 (2022).
- R20. Mañosa, L. *et al.* Inverse barocaloric effect in the giant magnetocaloric La–Fe–Si–Co compound. *Nat. Commun.* **2**, 595 (2011).
- R21. Strässle, Th., Furrer, A., Hossain, Z. & Geibel, Ch. Magnetic cooling by the application of external pressure in rare-earth compounds. *Phys. Rev. B* **67**, 054407 (2003).
- R22. Zhang, Z. *et al.* Thermal batteries based on inverse barocaloric effects. *Sci. Adv.* **9**, eadd0374 (2023).
- R23. Romanini, M. *et al.* Giant and reversible barocaloric effect in trinuclear spin-crossover complex Fe₃(bntz)₆(tcnset)₆. *Adv. Mater.* **33**, 2008076 (2021).
- R24. Aznar, A. *et al.* Reversible colossal barocaloric effects near room temperature in 1-X-adamantane (X=Cl, Br) plastic crystals. *Applied Materials Today* **23**, 101023 (2021).
- R25. Gao, Y. *et al.* Reversible colossal barocaloric effect dominated by disordering of organic chains in (CH₃–(CH₂)_{n–1}–NH₃)₂MnCl₄ single crystals. *NPG Asia Mater.* **14**, 34 (2022).
- R26. Li, J. *et al.* Colossal reversible barocaloric effects in layered hybrid perovskite (C₁₀H₂₁NH₃)₂MnCl₄ under low pressure near room temperature. *Adv. Funct. Mater.* **31**, 2105154 (2021).
- R27. Aznar, A. *et al.* Reversible and irreversible colossal barocaloric effects in plastic crystals. *J. Mater. Chem. A* **8**, 639–647 (2020).
- R28. Salvatori, A. *et al.* Colossal barocaloric effects in adamantane derivatives for thermal management. *APL Materials* **10**, 111117 (2022).
- R29. Seo, J., Braun, J. D., Dev, V. M. & Mason, J. A. Driving Barocaloric Effects in a Molecular Spin-Crossover Complex at Low Pressures. *J. Am. Chem. Soc.* **144**, 6493–6503 (2022).
- R30. Mañosa, L. & Planes, A. Solid-state cooling by stress: A perspective. *Appl. Phys. Lett.* **116**, 050501 (2020).
- R31. Li, B. *et al.* Colossal barocaloric effects in plastic crystals. *Nature* **567**, 506–510 (2019).
- R32. Lloveras, P. *et al.* Colossal barocaloric effects near room temperature in plastic crystals of neopentylglycol. *Nat. Commun.* **10**, 1803 (2019).
- R33. Tamarit, J. Li., Legendre, B. & Buisine, J. M. Thermodynamic study of some neopentane derivated by thermobarometric analysis. *Mol. Cryst. Liq. Cryst.* **250**, 347–358 (1994).
- R34. Eilerman, D. & Rudman, R. Polymorphism of crystalline poly(hydroxymethyl) compounds. III. The structures of crystalline and plastic tris(hydroxymethyl)aminomethane. *The Journal of Chemical Physics* **72**, 5656–5666 (1980).

We hope the revisions can satisfy the comments and suggestions from reviewers. See the highlighted by yellow. Any problems please contact me.

REVIEWER COMMENTS

Reviewer #1 (Remarks to the Author):

Since the authors have fully responded to my comments raised on the previous manuscript and accordingly updated the manuscript, I recommend the current version for publication in nature communications.

Reviewer #2 (Remarks to the Author):

The authors have improved the manuscript following the Reviewers comments. However, while I am happy with many of their responses, there are several ones that are not convincing. Please find my response below:

- Concerning my previous comment 2, the authors say that their compounds can be called plastic crystals because there is no need to exhibit a globular shape and the soft mechanical behavior and diffusion shown by their compound. Maybe my previous comment was misleading but, in addition to diffusion and soft mechanical behavior, “plastic crystals” are a class of orientationally disordered crystals whose entities are molecules that can rotate as a whole with respect to their centers of mass, and this is clearly not the case of the compound under study. That is, “plastic crystals” do not refer to crystals that are plastic as a unique condition.

- Concerning my previous comment 6, the authors say that higher rates could lead to artificial hysteresis due to measurement deficiencies, which they call extrinsic hysteresis. This is not convincing at all, as innumerable measurements using this technique show. In fact, the response of thermocouples or Pt-100 is fast enough to give accurate temperatures even at fast scanning rates of 20 K/min. As a couple of examples: (i) it is not a ubiquitous feature that phase transitions show larger hysteresis at larger scanning rates. (ii) very fast scanning rates may lead, in some cases, to glass phases, which are an example of an “infinite” hysteresis that is clearly intrinsic to the material. So the authors should treat their results at any scanning rate as intrinsic to the material.

- With my previous comment 10 concerning thermal expansion, I think that there has been a misunderstanding. I intended to mean that volume changes due to changes in temperature in each phase (hence not due to the phase transition) also lead to barocaloric effects that, for a proper determination of the barocaloric response, should be added to the response across the phase transition. This additional response can be determined using dV/dT in a Maxwell relation. As the authors have this quantity, then I suggest to add this contribution to the overall response.

- As a very minor comment, in the new version of the supplementary material some tables appear splitted in two pages. I think it would be much better to present each in a single page.

Dear reviewers,

We are grateful to you for the further comments and suggestions. The manuscript has been further improved accordingly.

Following are the point-by-point responses to the reviewers:

Reviewers' comments (Blue) and Responding to Reviewers' Comments (Black):

For Reviewer #1

Since the authors have fully responded to my comments raised on the previous manuscript and accordingly updated the manuscript, I recommend the current version for publication in nature communications.

Thanks so much for the recognition of the reviewer.

For Reviewer #2

The authors have improved the manuscript following the Reviewers comments. However, while I am happy with many of their responses, there are several ones that are not convincing. Please find my response below:

Thanks so much for the recognition of the reviewer. As for the responses which the reviewer thinks not convincing, the further responses and improvement of the manuscript will be shown below.

1. Concerning my previous comment 2, the authors say that their compounds can be called plastic crystals because there is no need to exhibit a globular shape and the soft mechanical behavior and diffusion shown by their compound. Maybe my previous comment was misleading but, in addition to diffusion and soft mechanical behavior, “plastic crystals” are a class of orientationally disordered crystals whose entities are molecules that can rotate as a whole with respect to their centers of mass, and this is clearly not the case of the compound under study. That is, “plastic crystals” do not refer to crystals that are plastic as a unique condition.

We appreciate the reminder of the reviewer with the precise attitude. We agree with the reviewer that the $(\text{CH}_3-(\text{CH}_2)_9)_2\text{NH}_2\text{Cl}$ (dC_{10}Cl for short) compound in our work presents the molecular structure without the typical spherical rotator like NPG. Although large molecules with disk-like

and rod-like (long alkyl chains for instance) species were considered to form plastic crystal phases in previous literatures [Laschat, S. *et al. Angew. Chem., Int. Ed.* **46**, 4832 (2007); Das, S. *et al. Chem. Soc. Rev.* **49**, 8878–8896 (2020)], to avoid the controversy of being plastic crystal, we choose to avoid the statement of dC₁₀Cl as plastic crystal in our manuscript according to the suggestions from the reviewer. For instance, we change the previous title “low pressure reversibly driving colossal barocaloric effect in two-dimensional vdW plastic crystals” to “low pressure reversibly driving colossal barocaloric effect in two-dimensional vdW alkylammonium halides”.

The corresponding revisions in the main text have been made, see the revised words marked in yellow. See Page 2 line 27, Page 4 line 81, Page 25 line 498, et al.

2. Concerning my previous comment 6, the authors say that higher rates could lead to artificial hysteresis due to measurement deficiencies, which they call extrinsic hysteresis. This is not convincing at all, as innumerable measurements using this technique show. In fact, the response of thermocouples or Pt-100 is fast enough to give accurate temperatures even at fast scanning rates of 20 K/min. As a couple of examples: (i) it is not a ubiquitous feature that phase transitions show larger hysteresis at larger scanning rates. (ii) very fast scanning rates may lead, in some cases, to glass phases, which are an example of an “infinite” hysteresis that is clearly intrinsic to the material. So the authors should treat their results at any scanning rate as intrinsic to the material.

We appreciate the comment of the considerate reviewer. On the suggestion here of the reviewer, we further consider and elucidate the relationship between the temperature scanning rate and the hysteresis of phase transition.

Hysteresis as the character of the first-order phase transition has been investigated in plentiful material systems, where the temperature scanning rate-dependence of the hysteresis were experimentally studied in RbNO₃ powder^{R1}, VO₂ ceramics^{R1}, FeMn alloys^{R1}, MnBi and MnTiBi intermetallic compounds^{R2}, and various spin-crossover compounds containing Co^{II}(dpzca)₂^{R3}, Fe^{II}₂(PMPhtBuT)₂(BF₄)₄·3.5H₂O^{R4} and FeL(HIm)₂^{R5}. There exists the kinetic effect of phase transition that the thermal hysteresis width performs an evident scan rate dependency in aforementioned systems, which is the larger hysteresis at larger scan rate more specifically. For instance, as shown in Fig. R1 and Fig. R2, the endothermal transition temperature increases and exothermal transition temperature decreases with enlarging scanning rate of temperature, therefore the hysteresis effect and consequent dissipation can be intensified at large scanning rate of temperature. Underlyingly, the phenomenon can be attributed to the less time left for thermal

activation to overcome the transition kinetic barrier in material systems at high scan rate of temperature, which causes the transition farther away from the equilibrium (i.e., higher for the heating process, lower for the cooling process and larger hysteresis)^{R2}.

Fig. R1 Heat flux q_s versus temperature T as a function of the temperature scan rate dT/dt from DSC scans in MnBi bulk samples^{R2}.

Fig. R2 The observed exothermal and endothermal phase transition temperature ($T_{1/2}$ cooling and $T_{1/2}$ warming respectively) values as a function of the scan rate (plotted on a semi-log scale) in the range 0.2 K min^{-1} to 10 K min^{-1} , for $[\text{Co}^{\text{II}}(\text{dpzca})_2]$ ^{R3}.

For our dC_{10}Cl compound, the rate scan-dependency of hysteresis, shown in Fig. R3, exhibits similar behavior that hysteresis increases from 5.9 K at T rate of 0.1 K/min to 8 K at T rate of 1 K/min, then to 10 K at T rate of 2 K/min, and we explain it with the thermally activated kinetics as well. Furthermore, at variable scan rates, the entropy changes of phase transition remain nearly

unchanged $\sim 400 \text{ J kg}^{-1} \text{ K}^{-1}$ on cooling and heating as shown in Fig. R4, which demonstrates the reversible and identical phase transition behavior at different temperature scanning rates.

Fig. R3 Heat flow of dC_{10}Cl at variable temperature rate of 0.1, 1 and 2 K/min.

Fig. R4 Entropy change of dC_{10}Cl based on the heat flow curves at variable temperature rates.

The inappropriate descriptions about extrinsic hysteresis in main text have been revised. The relevant description is modified to “noted that higher temperature rate induces thermally activated kinetics-related effect on hysteresis and consequent barocaloric effect; specific analysis can be seen in Supplementary Note 2”. See Page 9, line 187.

The effect of temperature rate on the BCE performance has also been objectively given in Supplementary Information (Supplementary Note 2) as below.

“we performed the DSC measurements at temperature rates of 0.1, 1, 2 K/min, and the appeared hysteresis is 5.9 K, 8 K, 10 K, respectively, while the phase transition entropy change remains constant ($\sim 400 \text{ J kg}^{-1} \text{ K}^{-1}$). As a result, the pressure driving the maximum reversible entropy change ($\sim 400 \text{ J kg}^{-1} \text{ K}^{-1}$) slightly enhanced to 0.1 GPa at temperature rate 2 K/min, noting this value is 0.08

GPa in the case of 1 K/min.”

The discussion above has been included in Supplementary Note 2. See Supplementary Information. And the relevant content has been revised in the manuscript. See Page 9, line 187.

3. With my previous comment 10 concerning thermal expansion, I think that there has been a misunderstanding. I intended to mean that volume changes due to changes in temperature in each phase (hence not due to the phase transition) also lead to barocaloric effects that, for a proper determination of the barocaloric response, should be added to the response across the phase transition. This additional response can be determined using dV/dT in a Maxwell relation. As the authors have this quantity, then I suggest to add this contribution to the overall response.

We appreciate the careful suggestion of the reviewer. Based on the temperature-volume curve shown in Fig. 5c of the manuscript, we obtain the change rate of unit cell volume with temperature (dV/dT) $\sim 0.5 \text{ E-30 m}^3 \text{ K}^{-1}$ at both low-temperature-state and high-temperature-state under atmospheric pressure. Assuming the independent dV/dT on pressure, the additional barocaloric effect ΔS^+ can be obtained with the Maxwell relation of $dV/dT = -dS/dP$. Therefore, with relation $[\Delta S^+(P) = -(dV/dT)_{p=0} * P]$, additional entropy change can be estimated as $-9 \text{ J kg}^{-1} \text{ K}^{-1}$ under pressurization of 20 MPa and $-45 \text{ J kg}^{-1} \text{ K}^{-1}$ under pressurization of 100 MPa, which emphasizes that the volume expansion in each phase contributes to the enhanced barocaloric effect.

The discussion about additional barocaloric effect has been included in Supplementary Note 4, as seen in the revised Supplementary Information.

4. As a very minor comment, in the new version of the supplementary material some tables appear splitted in two pages. I think it would be much better to present each in a single page.

Thanks for the suggestion of the reviewer. The tables in supplementary material have been adjusted and shown in a single page.

References:

- R1. Zhang, J., Zhong, F. & Siu, G. The scanning-rate dependence of energy dissipation in first-order phase transition of solids. *Solid State Communications* **97**, 847–850 (1996).
- R2. Basso, V., Piazzzi, M., Bennati, C. & Curcio, C. Hysteresis and phase transition kinetics in magnetocaloric materials. *Physica Status Solidi (b)* **255**, 1700278 (2018).
- R3. Miller, R. G., Narayanaswamy, S., Tallon, J. L. & Brooker, S. Spin crossover with thermal

- hysteresis in cobalt(ii) complexes and the importance of scan rate. *New J. Chem.* **38**, 1932 (2014).
- R4. Kulmaczewski, R. *et al.* Remarkable scan rate dependence for a highly constrained dinuclear iron(II) spin crossover complex with a wide thermal hysteresis loop. *J. Am. Chem. Soc.* **136**, 878–881 (2014).
- R5. Kurz, H., Hörner, G. & Weber, B. An iron(II) spin crossover complex with a Maleonitrile Schiff base-like ligand and scan rate-dependent hysteresis above room temperature. *Zeitschrift anorg allge chemie* **647**, 896–904 (2021).

We hope the revisions can satisfy the comments and suggestions from reviewers. See the highlighted by yellow. Any problems please contact me.

REVIEWER COMMENTS

Reviewer #2 (Remarks to the Author):

I am very sorry but I cannot give my approval to the new version either. Below my comments:

1) With respect to my previous with respect to the inadequacy of calling their compound “plastic crystal”, I am happy with their changes, so in my opinion no further changes are needed. However, just for completeness, I want to mention that the authors’ response is not satisfactory. The first paper cited in their response does not mention plastic crystals in any place, but disk- and rod-like shapes refer to liquid crystals. In the second paper I could only find text about rod-like shapes referring to “macroscopic shaping” of samples at a scale much larger than the molecular or the unit cell scale. But regardless of that, I want to remark here that the key point for the definition of plastic crystals do not deal with specific shapes but with the fact that the entities rotate as a whole. In fact, the second paper cited by the authors in their response states that “While PCs are defined by their whole molecular motions, the amphidynamic crystals comprise a static (stator) reference frame and a dynamic (rotator) frame in the constituent covalent or supramolecular unit (the combination is called a rotor)”.

2) With respect to the hysteresis issue, I cannot accept including such a couple of examples in their supplementary material. On the one hand, in the first example it is precisely shown that the onset transition temperature (the actual transition temperature for pure compounds undergoing isothermal transitions) of the endothermic transition is nearly independent of the scanning rate whereas for the exothermic transition it does depend on it. This precisely a counterexample that shows that the hysteresis is not a measurement effect (the shift of the maximum of the peak and the peak width are indeed a measurement effect but not the peak onset). Indeed, long history shows that typically endothermic transitions take place in equilibrium whereas exothermic transitions may not. On the other hand, in the second example the transition temperature for the SCO compound is determined at the midpoint of the magnetization increase, which is usually done by tradition coming from magnetic alloys undergoing athermal transitions (rather than isothermal), where the hysteresis is nearly independent of the scanning rate. This definition (which is highly disputable from a physical point of view) is in any case not the same as for the current compound for which the comparison is unfair and misleading. In addition to removing these plots, the authors should mention in some place that the hysteresis can be larger for larger scanning rates and that this could critically influence the reversibility of the BC effects depending on the operation conditions.

I understand that these criticisms do not deal with the main finding of the manuscript, however I think they should be addressed for the manuscript to be accepted for publication.

Dear reviewer,

We are grateful to you for the further comments and suggestions. The manuscript has been further improved accordingly.

Following are the point-by-point responses to the reviewer:

Reviewer's comments (Blue) and Responding to Reviewer's Comments (Black):

For Reviewer #2

I am very sorry but I cannot give my approval to the new version either. Below my comments:

We really appreciate the thorough thinking and patient guidance of the reviewer, and our responses can be seen below.

1. With respect to my previous with respect to the inadequacy of calling their compound “plastic crystal” , I am happy with their changes, so in my opinion no further changes are needed. However, just for completeness, I want to mention that the authors’ response is not satisfactory. The first paper cited in their response does not mention plastic crystals in any place, but disk- and rod-like shapes refer to liquid crystals. In the second paper I could only find text about rod-like shapes referring to “macroscopic shaping” of samples at a scale much larger than the molecular or the unit cell scale. But regardless of that, I want to remark here that the key point for the definition of plastic crystals do not deal with specific shapes but with the fact that the entities rotate as a whole. In fact, the second paper cited by the authors in their response states that “While PCs are defined by their whole molecular motions, the amphidynamic crystals comprise a static (stator) reference frame and a dynamic (rotator) frame in the constituent covalent or supramolecular unit (the combination is called a rotor)” .

We appreciate the insight of the reviewer into the key point for being plastic crystal. We agree with the reviewer that the plastic crystals should be defined by their whole molecular motions. With the key point mentioned by the reviewer, we further understand the plastic crystals and avoid the misconception about plastic crystals in future.

2. With respect to the hysteresis issue, I cannot accept including such a couple of examples in their supplementary material. On the one hand, in the first example it is precisely shown that the onset transition temperature (the actual transition temperature for pure compounds undergoing isothermal transitions) of the endothermic transition is nearly independent of the scanning rate whereas for the exothermic transition it does depend on it. This precisely a counterexample that shows that the hysteresis is not a measurement effect (the shift of the maximum of the peak and the peak width are indeed a measurement effect but not the peak onset). Indeed, long history shows that typically endothermic transitions take place in equilibrium whereas exothermic transitions may not. On the other hand, in the second example the transition temperature for the SCO compound is determined at the midpoint of the magnetization increase, which is usually done by tradition coming from magnetic alloys undergoing athermal transitions (rather than isothermal), where the hysteresis is nearly independent of the scanning rate. This definition (which is highly disputable from a physical point of view) is in any case not the same as for the current compound for which the comparison is

unfair and misleading. In addition to removing these plots, the authors should mention in some place that the hysteresis can be larger for larger scanning rates and that this could critically influence the reversibility of the BC effects depending on the operation conditions.

We appreciate the careful comments of the reviewer. With the detailed explanation given by the reviewer, we know that the temperature scanning rate-dependent hysteresis effect is not ubiquitous and exhibits variable behaviors (such as the different responses of endothermal and exothermal peaks) in variable material systems, which involves the intrinsic character of the compound, possible kinetic effect of the phase transition, and so on.

According to the suggestions, we have removed the inadequate description and the mentioned plots in the Supplementary Note 2. And we add the statement in the manuscript that “the hysteresis can be larger for higher temperature scanning rates, which could critically influence the reversibility of the BC effects depending on the operation conditions”. See Page 9 line 187 and Supplementary Note 2 in the Supplementary Information.

I understand that these criticisms do not deal with the main finding of the manuscript, however I think they should be addressed for the manuscript to be accepted for publication.

We appreciate the reviewer’s suggestions, which advance our manuscript a lot.

We hope our responses and revisions can satisfy the reviewer. See the revisions highlighted by yellow. Any problems please contact me.

REVIEWERS' COMMENTS

Reviewer #2 (Remarks to the Author):

After the new revision, I find that the manuscript deserves publication in Nature Communications as is.